# Polycystin-2 is an essential ion channel subunit in the primary cilium of the renal collecting duct epithelium

Xiaowen Liu[1,2†], Thuy Vien[3], Jingjing Duan[1,2‡], Shu-Hsien Sheu[1,2,4‡], Paul G DeCaen[3†*], David E Clapham[1,2‡*]

[1]Department of Cardiology, Howard Hughes Medical Institute, Boston Children's Hospital, Boston, United States; [2]Department of Neurobiology, Harvard Medical School, Boston, United States; [3]Department of Pharmacology, Northwestern University, Feinberg School of Medicine, Chicago, United States; [4]Department of Pathology, Boston Children's Hospital, Boston, United States

*For correspondence:
Paul.Decaen@northwestern.edu
(PGDC);
claphamd@hhmi.org (DEC)

†These authors contributed
equally to this work

Present address: ‡Janelia
Research Campus, Howard
Hughes Medical Institute,
Ashburn, United States

Competing interests: The
authors declare that no
competing interests exist.

Reviewing editor: Richard S
Lewis, Stanford University School
of Medicine, United States

**Abstract** Mutations in the polycystin genes, *PKD1* or *PKD2,* results in Autosomal Dominant Polycystic Kidney Disease (ADPKD). Although a genetic basis of ADPKD is established, we lack a clear understanding of polycystin proteins' functions as ion channels. This question remains unsolved largely because polycystins localize to the primary cilium – a tiny, antenna-like organelle. Using a new ADPKD mouse model, we observe primary cilia that are abnormally long in cells associated with cysts after conditional ablation of *Pkd1* or *Pkd2*. Using primary cultures of collecting duct cells, we show that polycystin-2, but not polycystin-1, is a required subunit for the ion channel in the primary cilium. The polycystin-2 channel preferentially conducts $K^+$ and $Na^+$; intraciliary $Ca^{2+}$, enhances its open probability. We introduce a novel method for measuring heterologous polycystin-2 channels in cilia, which will have utility in characterizing *PKD2* variants that cause ADPKD.
DOI: https://doi.org/10.7554/eLife.33183.001

## Introduction

Autosomal dominant polycystic kidney disease (ADPKD) is an adult-onset disease characterized by focal cyst development resulting from heterozygous mutations in *PKD1* or *PKD2* (*Brasier and Henske, 1997*; *Grantham, 2001*; *Hughes et al., 1995*; *Mochizuki et al., 1996*). While considered a dominant monogenic disease, the prevailing two-hit model states that ADPKD is recessive at the cellular level and that cysts develop from cells after acquiring a second somatic mutation to deactivate the remaining normal allele (*Koptides et al., 1999*; *Pei, 2001*; *Qian et al., 1997*; *Wu et al., 1998*). Mouse models of ADPKD implicate ciliary polycystin-1 and polycystin-2 dysfunction in kidney cyst formation. Complete genetic knockout of either *Pkd1* or *Pkd2* in mice results in embryonic lethality due to structural defects in the cardiovascular system, pancreas, and kidneys (*Boulter et al., 2001*; *Kim et al., 2000*; *Lu et al., 1997*; *Wu and Somlo, 2000*; *Wu et al., 2002*). The onset of kidney cyst development in adult mice following conditional inactivation of *Pkd1* or the intraflagellar transport protein kinesin, KIF3a (required for cilia formation), progresses well into adulthood, in analogy to the late progression of ADPKD in humans (*Davenport et al., 2007*; *Piontek et al., 2007*; *Shibazaki et al., 2008*). Conditional repression of either *Pkd1* (*Pax8^{rtTA}; TetO-cre; Pkd1^{fl/fl}*) (*Shibazaki et al., 2008*) or *Pkd2* (*Pax8^{rtTA}; TetO-cre; Pkd2^{fl/fl}*) (*Ma et al., 2013*) results in kidney cysts within 10 weeks after the start of doxycycline induction, suggesting that expression of both genes is necessary to prohibit cyst development in mature mice. Recently, the cystic phenotype found in mice deficient in *Pkd2* can be dose-dependently rescued by *Pkd2* transgene expression (*Li et al.,*

*2015*). Our poor understanding of the functional properties of polycystin-1, polycystin-2, and the polycystin-1/polycystin-2 complex impedes the development of therapeutic strategies; ADPKD is currently treated by dialysis and kidney transplant (*LaRiviere et al., 2015*). Since polycystin-2's function is unclear, it is not currently known if all ADPKD-causing variants in *Pkd2* cause a loss of function of the putative ion channel in primary cilia.

Polycystin-1 (PC1) is predicted to adopt an 11-transmembrane topology with a large autocleaved (G protein-coupled receptor proteolytic site, GPS) amino-terminal ectodomain (>3000 residues) (*Harris et al., 1995*) that is comprised of an array of putative adhesion and ligand-binding modules (*Burn et al., 1995*; *Hughes et al., 1995*; *Qian et al., 2002*). Polycystin-2 (or PC2, TRPP1, formerly TRPP2; encoded by *PKD2*) is a member of the large, 6-transmembrane spanning transient receptor potential (TRP) ion channel family (*Ramsey et al., 2006*; *Venkatachalam and Montell, 2007*) and has been observed to form a complex with polycystin-1 (*PKD1* gene). It is proposed to interact with polycystin-2 through a probable coiled-coil domain (*Newby et al., 2002*; *Qian et al., 1997*; *Tsiokas et al., 1997*), but in other experiments, the polycystin-1 and polycystin-2 interaction is preserved in overexpressed systems without the coiled-coil domain and is dependent on the N-terminal domain (*Babich et al., 2004*; *Ćelić et al., 2012*; *Feng et al., 2008*). However, the cryo-EM structure of purified polycystin-2 in lipid nanodiscs forms a homotetramer, with and without its C-terminal coiled-coil and N-terminal domains (*Shen et al., 2016*). Based on biochemistry and immunoreactivity, both proteins can be found in the primary cilium and ER (*Ong and Wheatley, 2003*; *Yoder et al., 2002*). In addition, some studies suggest that polycystin-1 and polycystin-2 may reciprocally affect each other's surface membrane or ciliary localization (*Harris et al., 1995*; *Ong and Wheatley, 2003*; *Xu et al., 2007a*). A recent study using inner medullary collecting duct (IMCD) cell lines derived from human ADPKD cysts suggests that impairing the function of polycystin-1 or polycystin-2 negatively affects the localization of the other polycystin: cells expressing an ADPKD-associated polycystin-1 mutation that prevents GPS domain cleavage have decreased amounts of both polycystin-1 and polycystin-2 in their primary cilia (*Xu et al., 2007a*). Our understanding of ADPKD pathogenesis is hampered by disagreements about the basic properties of the putative polycystin-2 current. Furthermore, little is understood regarding what role, if any, the primary cilia have in controlling the progression of cyst formation in ADPKD. Nonetheless, there is no ambiguity in the finding that mutations in *PKD1* or *PKD2* are genetically linked to formation of cysts in kidney and other tissues to cause significant morbidity and mortality in humans (*Mochizuki et al., 1996*; *Ong and Harris, 2015*).

Previous work reported single channel events from exogenously expressed polycystin-1 and polycystin-2 from the plasma membrane and in reconstituted polycystin-1 and polycystin-2 proteins recorded in formulated bilayers were attributed to an polycystin-1/polycystin-2 ion channel complex (*Delmas et al., 2004*; *González-Perrett et al., 2001*; *Hanaoka et al., 2000*). However, these non-ciliary preparations produced contradictory findings regarding its ion selectivity and voltage dependence: the polycystin-2 ion channel was initially reported to conduct calcium (*González-Perrett et al., 2001*; *Hanaoka et al., 2000*), and then not conduct calcium (*Cai et al., 2004*). Most recently, a gain-of-function mutation (F604P), but not *wt* polycystin-2, underlies a measurable current when heterologously expressed in *Xenopus* oocytes (*Arif Pavel et al., 2016*). This study demonstrated that plasma membrane polycystin-2 expression does not appear to be hampered by the lack of polycystin-1, but rather that native polycystin-2 channels appears to be constitutively closed unless mutated (F604P; affecting flexibility of the S5 segment as in TRPML1). The monovalent-selective current of this mutant is blocked by divalent ions ($Ca^{2+}$ and $Mg^{2+}$). As we will review in the Discussion, several putative polycystin-2 activators have been reported to sensitize polycystin-2 channels, but we have been unable to reproduce these results (*Kim et al., 2016*; *Leuenroth et al., 2007*). While the apparent differences observed can be rooted in methodology, these preparations are measured from non-ciliary membranes and thus share the same disadvantage. Recently, two methods have been used to measure ion currents from intact primary cilia (*DeCaen et al., 2013*; *Kleene and Kleene, 2012*), thus preserving their unique native microenvironment without the need for reconstitution.

For this study, we crossed our *Arl13b-EGFP^tg* strain (*DeCaen et al., 2013*) with *Pax8^rtTA*; *TetO-cre*; *Pkd1^fl/fl* (cPkd1) or *Pax8^rtTA*; *TetO-cre*; *Pkd2^fl/fl* (cPkd2) mice provided by the Somlo lab (*Ma et al., 2013*; *Shibazaki et al., 2008*). The progeny express the *Arl13b-EGFP^tg* cilia reporter, and ablation of either *Pkd1* or *Pkd2* genes expression in the kidney under the Pax8^rtTA

promotor (*Traykova-Brauch et al., 2008*) is *TetO-cre* doxycycline-dependent. For brevity, we will call these animal strains either *Arl13b-EGFP^{tg}:cPkd1* or *Arl13b-EGFP^{tg}:cPkd2*, respectively. Consistent with previous reports (*Ma et al., 2013*), we find that repression of either *Pkd1* or *Pkd2* results in obvious kidney cysts within two months after removal of doxycycline. Primary cilia from cysts of either doxycycline-treated *Arl13b-EGFP^{tg}:cPkd1* or *Arl13b-EGFP^{tg}:cPkd2* mice were substantially elongated compared to control littermates. We utilized the cilium patch method to directly measure ciliary ion channels from primary cultures of inner medullary collecting duct epithelial cells (pIMCD) from these mice. We characterize the native ciliary polycystin-2 currents, which can be conditionally ablated using doxycycline in the *Arl13b-EGFP^{tg}:cPkd2* mouse model.

Surprisingly, polycystin-2 forms a functional ion channel in primary cilia without polycystin-1 expression, calling into question the hypothesis that polycystin-1 is an obligate subunit of putative polycystin-1/polycystin-2 heteromeric channel complex. The ciliary polycystin-2 current preferentially conducts the monovalents $K^+$ and $Na^+$, over divalent $Ca^{2+}$ ions. Millimolar external $[Ca^{2+}]$ weakly permeates through the polycystin-2 pore and blocks the inward sodium current. The open probability of polycystin-2 is enhanced by internal calcium ($EC_{50}$ = 1.3 µM), slightly exceeding the resting cilioplasmic $[Ca^{2+}]$ (~300–600 nM) (*Delling et al., 2013*, *2016*). Native constitutive plasma membrane currents are not affected by conditional ablation of either *Pkd1* or *Pkd2* from pIMCD cells. Thus, we find no evidence for homomeric or heteromeric polycystin channels in the plasma membrane. Heterologous, stably-expressed polycystin-2-GFP traffics to the primary cilia of HEK-293 cells, where cilia patch clamp recordings recapitulate the ion selectivity and internal calcium potentiation effects observed in primary cilia of native pIMCD cells.

## Results

### Progressive cyst formation in a new mouse model

Previous work demonstrated that the human ADPKD kidney cyst phenotype can be reproduced in mice 14 weeks after conditional ablation of nephron-localized *Pkd1* or *Pkd2* (*Ma et al., 2013*). To understand the putative ciliary ion channel function of polycystin-1 and/or polycystin-2, and to determine the effects of kidney cyst formation on cilia morphology, we crossed our *Arl13b-EGFP^{tg}* strain (*DeCaen et al., 2013*) with *cPkd1* or *cPkd2* mice (provided by S. Somlo Yale Univ.). We then induced either *Pkd1* or *Pkd2* gene inactivation in adult animals (~P28) by introducing doxycycline (2 mg/ml or 3.9 mM) into the drinking water for two weeks. After this treatment period, doxycycline was removed and kidney histology was performed from 2 and 4 month post-treatment animals (*Figure 1A*, *Figure 1—figure supplement 1*, *Figure 1—figure supplement 2A,B*).

Immunoblots were performed from pIMCD cell lysates from 2 week and 2 month doxycycline-ablated *Arl13b-EGFP^{tg}:cPkd2* or 2 week post-treatment of *Arl13b-EGFP^{tg}:cPkd1* mice, indicating that the recombinase substantially reduced polycystin-2 or polycystin-1 protein expression (*Figure 1B*). Polycystin-2 expression from *Arl13b-EGFP^{tg}:cPkd1* animals was unaffected by polycystin-1 ablation (*Figure 1B*). Consistent with previous reports from the *cPkd2 and cPkd1* strain, we observed kidney cyst formation in *Arl13b-EGFP^{tg}:cPkd2* and *Arl13b-EGFP^{tg}:cPkd1* mice (*Figure 1—figure supplement 1*, *Figure 1—figure supplement 2B*). The extent of cyst formation in these mice was quantified as the kidney-to-body weight ratio and cystic index (*Figure 1C–F*). Based on these measures, the cystic phenotype was progressive, as seen by comparing the post 2 month and 4 month treatment groups (*Figure 1C–F*, *Figure 1—figure supplement 1*, *Figure 1—figure supplement 2B*).

### Abnormal cilia in *Pkd1*- or *Pkd2*-ablated mice

Using confocal microscopy, we compared cilia morphology from kidneys of *Arl13b-EGFP^{tg}:cPkd2* and *Arl13b-EGFP^{tg}:cPkd1* mice treated with or without doxycycline (*Figure 2A–C*, *Figure 1—figure supplement 2C–E*). Here, we observed an ~3.2 fold increase in cilia length with the progression of ADPKD (5.7 ± 0.4 µm for 2 months and 18.4 ± 1.2 µm for 4 months post-treatment) with *Arl13b-EGFP^{tg}:cPkd2* mice, whereas cilia length from control littermates did not differ substantially over the same time course (3.8 ± 0.16 µm and 4.5 ± 0.2 µm, respectively). Also, we found that cilia length from tubule cells lining cysts were ~4 times longer than from unaffected tubules from the same animals (*Figure 2C*) (12 ± 1.1 µm and 3.1 ± 0.2 µm, respectively). As for *Arl13b-EGFP^{tg}:cPkd1* mice, we

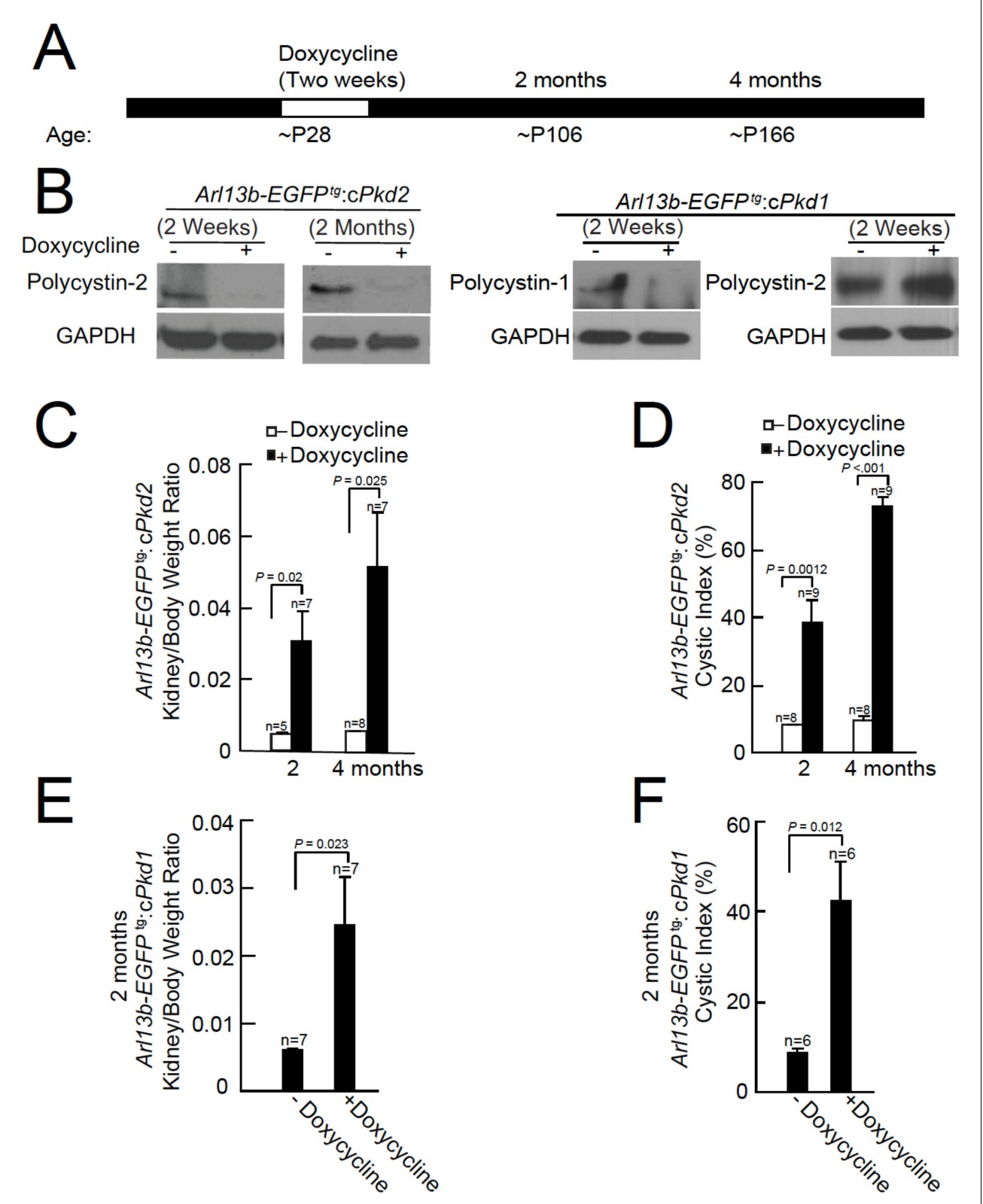

**Figure 1.** Onset of kidney tubule cyst formation in *Arl13b-EGFP^tg:cPkd1* and *Arl13b-EGFP^tg:cPkd2* animals. (**A**) Study design to assess cyst formation after genetic ablation of either *Pkd1* or *Pkd2*. One protocol is shown that assesses polycystin-1 or polycystin-2 at P28 in the conditional knockout mouse. (**B**) Loss of polycystin-1 and polycystin-2 protein expression as assessed by immunoblot, 2 weeks and 2 months after doxycycline removal. Whole cell lysates were prepared from pIMCD cells and subjected to western blot analysis (three independent experiments). A trace amount of
*Figure 1 continued on next page*

*Figure 1 continued*

polycystin-2 (first row, faint band at 2 weeks after doxycycline removal) is similar to that previously reported (*Ma et al., 2013*). The band at the left edge (*cPkd1*, 2 weeks + Doxycycline, first row), appears to be nonspecific, although we cannot rule out polycystin-1 contamination from non-tubule cells. (C) Kidney weight/body was increased in *Arl13b-EGFP^tg:cPkd2* mice with doxycycline treatment compared to control littermates without doxycycline treatment. (D) Cystic index (Materials and methods) shows that cysts increased in *Arl13b-EGFP^tg:cPkd2* mice. (E) Kidney weight/body was increased in *Arl13b-EGFP^tg:cPkd1* with doxycycline treatment (2 months after doxycycline removal) compared to control littermates without doxycycline treatment. (F) Cystic index shows increased size and number of cysts in *Arl13b-EGFP^tg:cPkd1* mice.

DOI: https://doi.org/10.7554/eLife.33183.002

The following figure supplements are available for figure 1:

**Figure supplement 1.** Progression of ADPKD after genetic ablation of *Pkd2*.
DOI: https://doi.org/10.7554/eLife.33183.003

**Figure supplement 2.** Progression of ADPKD after genetic ablation of *Pkd1*.
DOI: https://doi.org/10.7554/eLife.33183.004

observed an ~2.4 fold increase in cilia length with the progression of ADPKD (4.1 ± 0.1 µm for control littermates and 9.9 ± 0.5 µm for 2 months post-treatment)(*Figure 1—figure supplement 2D*). These results demonstrate the neither polycystin-1 nor polycystin-2 expression is required for primary ciliogenesis from the tubule epithelium, but implies that polycystin-1 or polycystin-2 expression is somehow related to cilia length. Since aberrant cilia morphology was mostly found in cystic tissue epithelia compared to non-cystic tubules, ciliary polycystin-1 or polycystin-2 may regulate continuing renal tubular cell differentiation. However, it is unclear if irregular cilia morphology is a consequence or cause of cyst formation, and what function overexpression of *Arl13b-EGFP* in combination with polycystin-1 or polycystin-2 ablation may have in maintaining normal cilia length.

## Ciliary trafficking and ion channel activity of polycystin-2 are independent of polycystin-1

Using animals from the same study design, we harvested pIMCD cells from 2-month-old *Arl13b-EGFP^tg* mice, before cyst development. The cell membrane of the dissociated cells retained anti-aquaporin 2 antibody reactivity and Arl13B was found in the primary cilia of intact distal collecting ducts (*Figure 3A–C*). Using the validated antibody described in *Figure 1*, we confirmed the lack of ciliary polycystin-2 from cultured pIMCD cells from post-doxycycline-treated *Arl13b-EGFP^tg:cPkd2* mice (*Figure 3D*). Importantly, the pIMCD cells isolated from post-doxycycline-treated *Arl13b-EGFP^tg:cPkd1* animals retained their ciliary polycystin-2, suggesting that ciliary polycystin-2 trafficking does not require polycystin-1 (*Figure 3E*, *Figure 1—figure supplement 2E*).

Next, we patch clamped pIMCD cells, in which the primary cilia could be visualized and expression of either polycystin-1 or polycystin-2 subunits of the putative ciliary ion channel complex could be conditionally controlled. Previously, we used the cilium patch method to identify the heteromeric polycystin 1-L1/polycystin 2-L1 channel in primary cilium of *Arl13b-EGFP^tg* retinal pigmented epithelial cells (RPE) and mouse embryonic fibroblasts (MEF) (*DeCaen et al., 2013*). Also, we described, but did not identify, a large outward conductance channel (outward γ = 98 ± 2 pS) from the cilia of an immortalized IMCD-3 cell line, which has been characterized (outward γ = 96 pS) and subsequently identified as polycystin-2 by the Kleene group (*Kleene and Kleene, 2017*). Thus, we extended our cilia electrophysiology methods to test ciliary ion currents from pIMCD cells and determine if polycystin-1 and/or polycystin-2 are subunits of the ion channel. After establishing high resistance seals (>16 GΩ) at the tip of the cilia membrane (*Video 1*), we ruptured the cilium's membrane and established 'whole-cilium' patch recording to observe an outwardly rectifying current (*Figure 4A*, *Figure 4—source data 1*-ciliary current amplitudes: siRNA screen of TRP proteins in cilia).

To determine the identity of this current, we treated cells with siRNA specific for members of the polycystin family and other localized putative ciliary ion channel subunits (*Kleene and Kleene, 2017*; *Köttgen et al., 2008*; *Yoder et al., 2002*). We observed 53% and 61% attenuation of whole-cilium current from cells treated with two independent siRNAs targeted to polycystin-2 (*Figure 4A*, *Table 1*, *Figure 4—source data 1*-ciliary current amplitudes: siRNA screen of TRP proteins in cilia). Importantly, we did not find any difference in currents when cells were treated with siRNAs targeting *Pkd1*, *Pkd1-1L1*, *Pkd2-L1*, and *Trpv4*, suggesting that none of these targets are essential subunits of

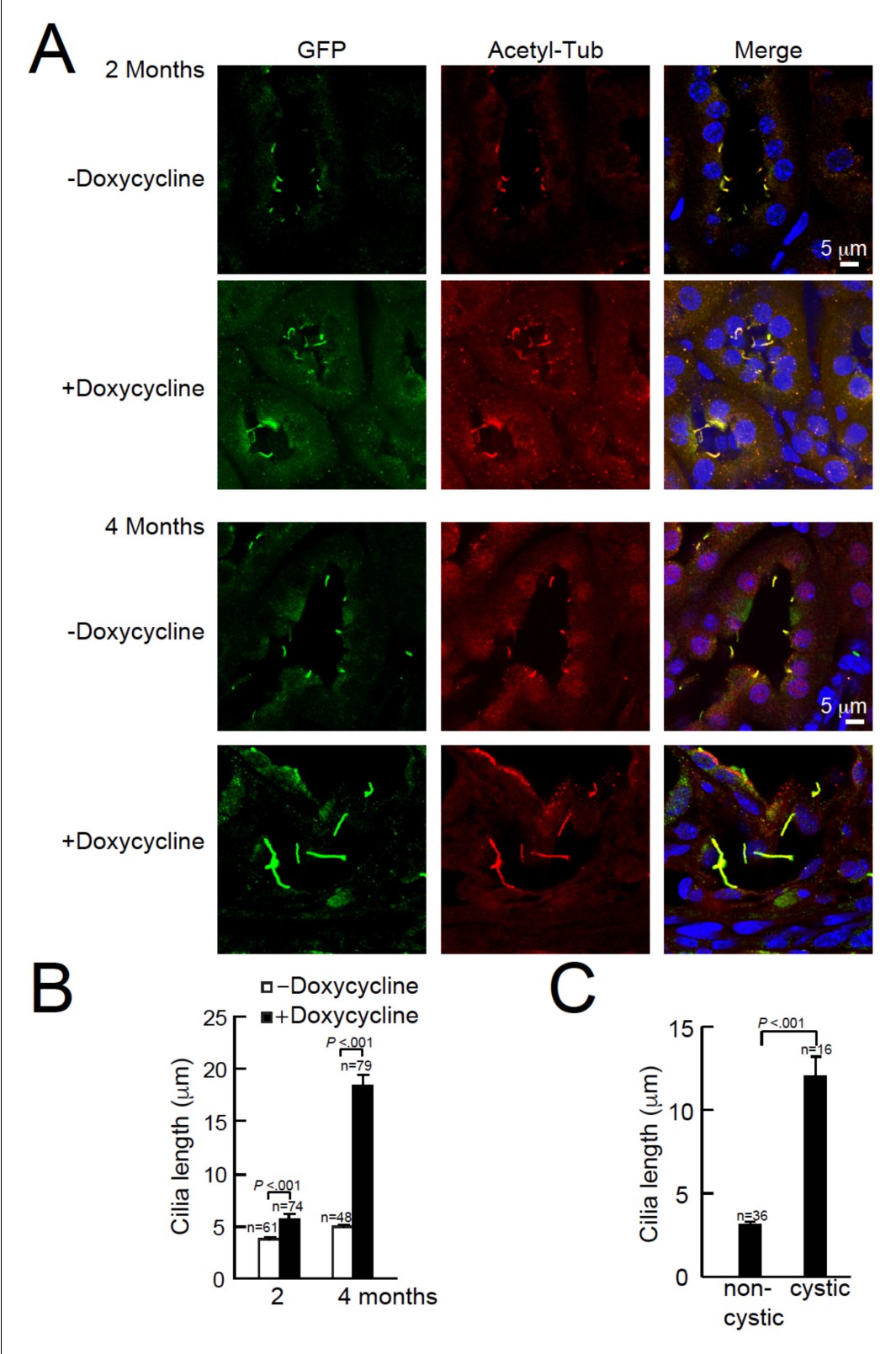

**Figure 2.** Cystic kidney cilia are abnormally long compared to unaffected tubules after genetic ablation of *Pkd2*. (**A**) Representative kidney sections from *Arl13b-EGFP^tg:cPkd2* mice were immunolabeled with antibodies against EGFP and acetylated tubulin. Three independent experiments were performed. Scale bars, 5 μm. (**B**) Cilia length was measured with the progression of cyst formation from kidney of *Arl13b-EGFP^tg:cPkd2* mice. (**C**) Cilia length was measured in cystic and non-cystic areas from the *Arl13b-EGFP^tg:cPkd2* mice after 2 months of doxycycline removal.
*Figure 2 continued on next page*

*Figure 2 continued*

DOI: https://doi.org/10.7554/eLife.33183.005

the pIMCD ciliary current. To confirm these results, we measured ciliary current of pIMCD cells from *Arl13b-EGFP*[tg]*:cPkd2* mice at 2 weeks and 2 months after withdrawal of doxycycline treatment. As expected, the ciliary outwardly rectifying currents from the *Arl13b-EGFP*[tg]*:cPkd2* mice were reduced by 84% and 81% from 2 week and 2 month post-treatment groups compared to littermates not exposed to doxycycline (*Figure 4B*, *Figure 4—source data 2*-whole ciliary current amplitudes in *Pkd1* or *Pkd2*-knockout primary cells). These results demonstrate that doxycycline-induced TetO-cre ablation of *Pkd2* substantially reduces the pIMCD ciliary current. In contrast, cilia currents recorded from pIMCD cells isolated from doxycycline-treated *Arl13b-EGFP*[tg]*:cPkd1* mice do not have reduced current compared to cells from untreated animals from 2 week and 2 month post-treatment groups (*Figure 4B*, *Figure 4—source data 2*-whole ciliary current amplitudes in *Pkd1* or *Pkd2*-knockout primary cells). From this data, we conclude that polycystin-2 is a subunit of a major ion current in renal tubule epithelial cilia and that the absence of polycystin-1 expression does not substantially alter the net polycystin-2 current in the cilium.

## Ciliary polycystin-2 preferentially conducts K$^+$ and Na$^+$ over Ca$^{2+}$ ions

As discussed in the introduction, it is widely reported that calcium is a major charge carrier for polycystin-2 under physiological conditions. However, we find that the collecting duct epithelial cilia membrane is ~2.5 times more selective for potassium than sodium ions (relative permeability $P_K$/$P_{Na}$ = 2.4, *Figure 5—figure supplement 1A*). Here, the relative permeability was estimated by the measured change in reversal potential when sodium was replaced by each test cation (*Table 2*). To test calcium permeability, the 110 mM NaCl extracellular solution was replaced with equimolar CaCl$_2$ (keeping 110 mM internal Na$^+$) which negatively shifted the reversal potential ($\Delta E_{rev}$ = −57 mV, *Table 2*), indicating that permeation by calcium ($P_{Ca}$/$P_{Na}$ = 0.06) is barely different than presumably impermeant NMDG ($P_{NMDG}$/$P_{Na}$ = 0.04). We also tested the permeability of chloride ($P_{Cl^-}$) by substituting external Cl$^-$ with the larger methane sulfonate while keeping [Na$^+$] constant (110 mM NaCl vs. NaMES). Here, there was no difference in reversal potential when NaCl was exchanged for NaMES ($\Delta E_{rev}$ = −2 ± 3 mV, *Table 2*), demonstrating that $P_{Cl^-}$ is negligible in the pIMCD cilium membrane. These data also demonstrate that polycystin-2's selectivity is strictly cationic and distinct from that previously reported for polycystin 1-L1/2-L1 recorded in the cilia of RPE and MEF cells, which was ~6 x more selective for Ca$^{2+}$ over Na$^+$ and K$^+$ (*DeCaen et al., 2013*).

We tested the effect of changing external calcium ([Ca$^{2+}$]$_{ex}$) while maintaining a constant level of Na$^+$ (100 mM) on the magnitude on the inward ciliary current (*Figure 5—figure supplement 1B*). Here we observed that the inward current, presumably carried by Na$^+$, was antagonized by [Ca$^{2+}$]$_{ex}$ (IC$_{50}$ = 17 mM). This appears to be a consistent feature of polycystin-2 and mutated forms of the polycystin-2 channels when recorded from oocytes and reconstitution preparations (*Arif Pavel et al., 2016*; *Cai et al., 2004*; *Koulen et al., 2002*; *Vassilev et al., 2001*). To validate our findings of ciliary relative permeability, we also compared the single channel conductance of inward Na$^+$, K$^+$, and Ca$^{2+}$ when they were exclusively present in the pipette (cilium-attached configuration). Of the three ions tested, K$^+$ conducted through ciliary polycystin-2 channels with the greatest inward conductance ($\gamma_K$ = 144 ± 6 pS), followed by sodium ($\gamma_{Na}$ = 89 ± 4 pS) and calcium ($\gamma_{Ca}$ = 4 ± 2 pS) (*Figure 5*). The inward Ca$^{2+}$ single channel currents were only observed under high electrical driving forces – when the ciliary membrane was hyperpolarized more negative than −140 mV (*Figure 5B,C*). Note that the outward conductance for all three conditions ranged between 90–117 pS, suggesting that the outward conductance is likely a mixture of Na$^+$ and K$^+$ exiting the cilium. The inward single channel open events were brief, usually lasting less than 0.5 ms (I$_{Na}$ open time 0.4 ± 0.2 ms at −100 mV), whereas those measured at positive potentials opened for 190 times longer (I$_{Na}$ open time 76 ± 29 ms at 100 mV). Importantly, the inward and outward conductance were absent from pIMCD cilia patches measured from doxycycline-treated *Arl13b-EGFP*[tg]*:cPkd2* animals when Ca$^{2+}$ was used in the pipette (*Figure 5—figure supplement 2A–C*), demonstrating that the conductance (both inward and outward) is dependent on *Pkd2* expression. When we compare the extrapolated $\Delta E_{rev}$ (−61 mV) from single channel Ca$^{2+}$ and monovalent currents, we observe that $P_{Ca}$/$P_{mono}$ = 0.04

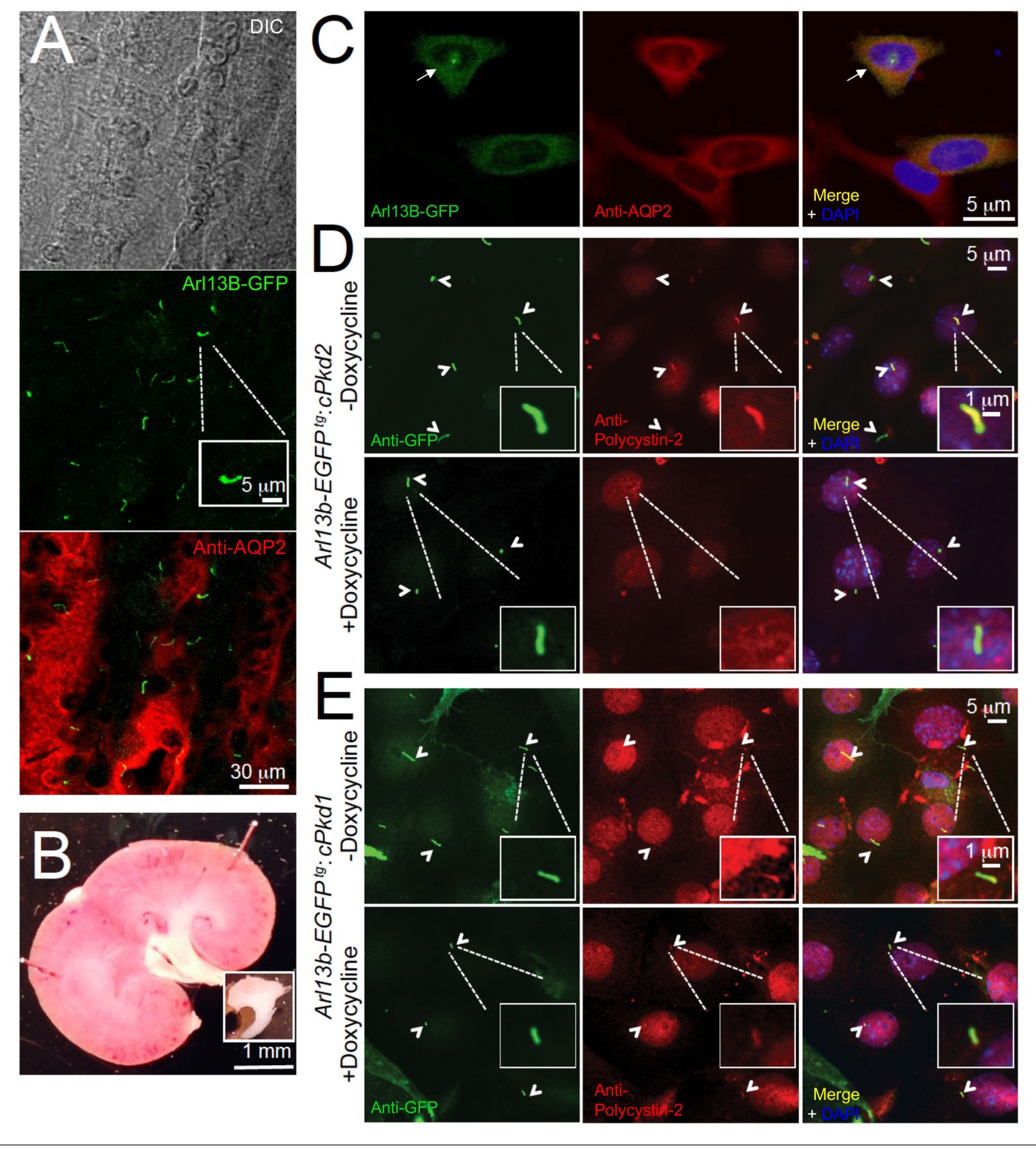

**Figure 3.** In situ and in vitro detection of ciliary polycystin-2 in *Arl13b-EGFP^tg* pIMCD cells. (**A**) Confocal images from a 100 μm-thick fixed kidney slice: DIC image in grey; aquaporin 2 (kidney collecting duct epithelial cell epitope) labeled with Alexa-569 (red); Cilia *Arl13b-EGFP^tg* (green). Three independent experiments were performed. Scale bar = 30 μm, inset image scale bar = 5 μm. (**B**) Sagittal section of 3-month-old mouse kidney with the inner medulla removed (bottom inset). Three independent experiments were performed. Scale bar = 1 mm. (**C**) Confocal images of fixed primary collecting duct epithelial cells after two days in culture, immunostained with anti-aquaporin 2 antibody in A). Three independent experiments were

*Figure 3 continued on next page*

*Figure 3 continued*

performed. Scale bar = 5 µm. (D) Immunofluorescence using anti-GFP (green) and anti-polycystin-2 (red) showing the loss of polycystin-2 in pIMCD cells isolated from kidney papillae of *Arl13b-EGFP^tg:cPkd2* mice (2 weeks after doxycycline removal; three independent experiments; 5 mice were used for each group). Arrowheads point to primary cilia. Scale bar = 5 µM, inset image scale bar = 1 µm. (E) Immunofluorescence with anti-GFP (green) and anti-polycystin-2 (red), showing ciliary polycystin-2 in pIMCD cells isolated from kidney papillae of *Arl13b-EGFP^tg:cPkd1* mice. Three independent experiments; 5 mice for each group. Arrowheads point to primary cilia. Scale bar = 5 µM, inset image scale bar = 1 µm.

DOI: https://doi.org/10.7554/eLife.33183.006

when we assume that [Ca$_{cilia}$] is high (580 nM) (*Delling et al., 2013*) and the cumulative ciliary monovalent concentration (155 mM) is similar to the cytosol (*Figure 5—figure supplement 2D*). Thus, the relative permeability of ion conductance dependent on polycystin-2 expression agrees with the relative permeability measured from the cilia membrane and the polycystin-2 single channels currents. We conclude that the major polycystin-2 conductance is monovalent, with relatively little inward Ca$^{2+}$ flux.

## Ciliary polycystin-2 is sensitized by intraciliary free calcium

Since intracellular calcium has been reported to sensitize polycystin-2 from the ER and cilia of cell lines (*Cai et al., 2004*; *Kleene and Kleene, 2017*) and the polycystin-like channel (also called polycystin -L or polycystin 2-L1) (*DeCaen et al., 2016*), we examined this property in pIMCD ciliary polycystin-2 channels. Inside-out cilia membrane single channel activity can be compared to varying levels of intraciliary free calcium ([free Ca$^{2+}$]$_{in}$) (*Figure 6A*). Most commonly, inside-out ciliary patches exhibited at least 3–4 active polycystin-2 channels, but some had only one polycystin-2 channel present (*Figure 6B*). We used these rare patches to determine that 3 µM [free Ca$^{2+}$]$_{in}$ enhanced the open probability of the polycystin-2 current ~10 times (increasing P$_o$ from 0.034 ± 0.02 to 0.36 ± 0.07) and the mean open time ~6 times (increasing from 37 ± 26 ms to 215 ± 40 ms) compared to standard cytoplasmic concentrations of 90 nM [free Ca$^{2+}$]$_{in}$ (*Figure 6B–D*). The half maximal enhancement of IMCD polycystin-2 open time was 1.3 µM. Previously, internal ciliary calcium was shown to negatively shift the voltage dependence of polycystin-2 channel activation in IMCD-3 cell lines (*Kleene and Kleene, 2017*). To determine internal calcium's effect of on polycystin-2 inward current at the cilia's resting membrane (RMP$_{cilia}$ = −18 mV) (*Delling et al., 2013*), we compared polycystin-2 single channels as we increased [free Ca$^{2+}$]$_{in}$. First, we measured single currents activated by voltage ramps (−100 to 100 mV) and subtracted the remaining ohmic current after channel inactivation (*Figure 6—figure supplement 1A*). At 600 nM [free Ca$^{2+}$]$_{in}$, the polycystin-2 channel typically remains closed at negative potentials (*Figure 6—figure supplement 1B*). However, when [free Ca$^{2+}$]$_{in}$ was increased to 30 µM, the open events were more frequent, which increased in the total inward current ~10 fold (summing the single channel events; −13 pA to −122 pA at RMP$_{cilia}$, *Figure 6—figure supplement 1B,C*). Similar to the observations made from the cilia current recorded from IMCD-3 cilia, increasing [free Ca$^{2+}$]$_{in}$ 50-fold substantially shifts the voltage

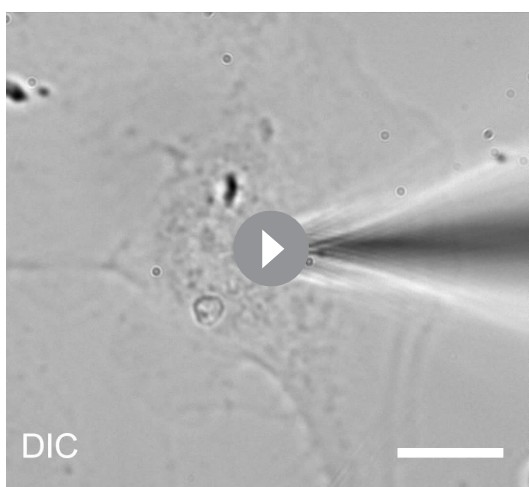

**Video 1.** Movie of the pIMCD *Arl13b* cilium patch configuration. The glass pipette patch electrode (*right*) is sealed onto a primary cilium above the pIMCD cell. The focal plane was moved along the z-axis (~9 µm) to visualize the cell and cilium. The electrode is moved along the y-axis while adjusting the focal plane to demonstrate that the patch electrode is sealed on the tip of the cilia membrane, not the cell membrane. The light source(s) are indicated in the lower left corner of the image: white light and 488 nm light to illuminate the specimen. Differential interference contrast (DIC) or fluorescent images were captured using a Hamamatsu Orca Flash CCD camera on an Olympus IX73 inverted microscope; 60x objective, 2x photomultiplier. Scale bar = 5 µm.

DOI: https://doi.org/10.7554/eLife.33183.010

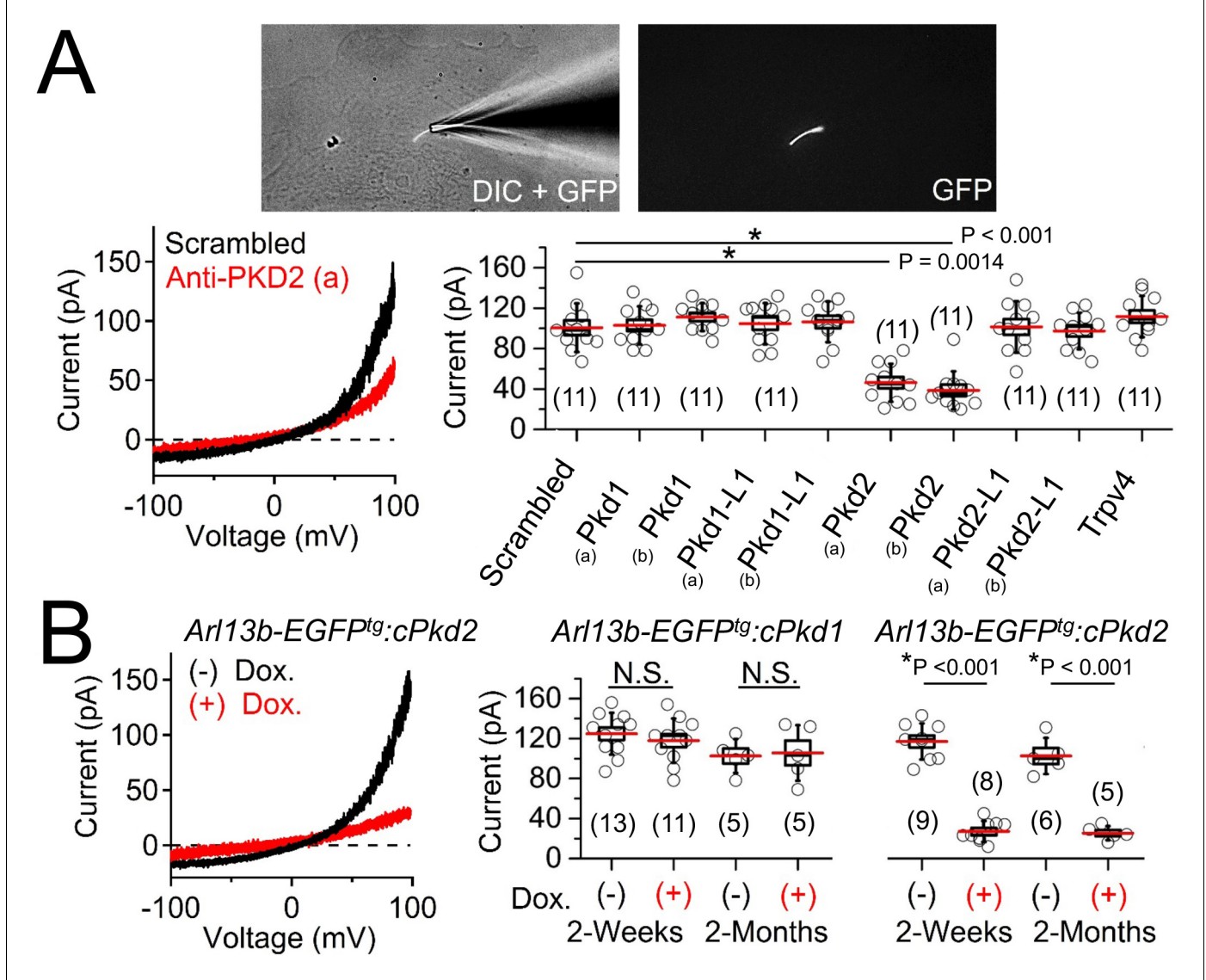

**Figure 4.** Polycystin-2 is required for the ciliary ion channel conductance of the primary inner medullary collecting duct epithelial cells (pIMCDs). (A) siRNA screen of potential $I_{cilia}$ candidates. *Top*, light microscope image of a patched cilium. *Left*, example ciliary currents measured from cells treated with either scrambled siRNA or one targeted to polycystin-2. *Right*, box (mean ± S.E.M.) and whisker (mean ± S.D.) plots of cilia total outward current (+100 mV) measured 48–72 hr after double-siRNA treatment. *Pkd1, Pkd1-L1, Pkd2, and Pkd2-L1* mRNAs were targeted by two siRNAs specific for two different regions (A, B) of the target transcript (listed in *Table 1*). Averages are indicated by the red lines. Student's *t*-test P values comparing treatment groups to scrambled siRNA. See *Figure 4—source data 1*-ciliary current amplitudes: siRNA screen of TRP proteins in cilia. (B) Conditional knockout of the whole-cilia current. *Left*, exemplar cilia currents from pIMCD epithelial cells isolated from conditional *Pkd2* knockout (*Arl13b-EGFP^tg:cPkd2*) transgenic mice. *Right*, box and whisker plots comparing the total outward cilia current (+100 mV) from control littermates and doxycycline-treated animals (*Arl13b-EGFP^tg:cPkd1* and *Arl13b-EGFP^tg:cPkd2*). Number of cilia; italic numeral in parentheses for each group genotype and treatment group. Student's *t*-test P values compare the outward cilia current from the untreated and doxycycline-treated animals. See *Figure 4—source data 2*- whole ciliary current amplitudes in *Pkd1* or *Pkd2*-knockout primary cells.

DOI: https://doi.org/10.7554/eLife.33183.007

The following source data is available for figure 4:

**Source data 1.** Ciliary current amplitudes: siRNA screen of TRP proteins in cilia.
DOI: https://doi.org/10.7554/eLife.33183.008
**Source data 2.** Whole ciliary current amplitudes in *Pkd1* or *Pkd2*-knockout primary cells.
DOI: https://doi.org/10.7554/eLife.33183.009

**Table 1 .** siRNAs used to screen for ciliary ion channel genes.

| siRNA gene target M, *Mus musculus*; h, *human* | ThermoFisher Silencer ID | siRNA location | % Knockdown efficiency |
|---|---|---|---|
| m*Pkd1* (a) | 151949 | 999 | 75 ± 3 |
| m*Pkd1* (b) | 151951 | 7303 | 73 ± 3 |
| m*Pkd1-L1* (a) | n398242 | 1075 | 76 ± 2 |
| m*Pkd1-L1* (b) | n398244 | 806 | 72 ± 3 |
| m*Pkd2* (a) | 150154 | 1089 | 83 ± 2 |
| m*Pkd2* (b) | 63551 | 488 | 81 ± 2 |
| m*Pkd2-L1* (a) | 101318 | 350 | 84 ± 2 |
| m*Pkd2-L1* (b) | 101422 | 830 | 82 ± 2 |
| m*Trpv4* | 182203 | 778 | 80 ± 2 |
| hPKD2 (a) | 104317 | 1100 | 74 ± 2 |
| hPKD2 (b) | 143288 | 3069 | 71 ± 3 |

DOI: https://doi.org/10.7554/eLife.33183.011

dependence of activation; *Figure 6—figure supplement 1D*). We also observed a ~ 10 x increase in the normalized conductance at $RMP_{cilia}$ (0.05 to 0.49), which was also observed at more negative potentials. In summary, ciliary calcium enhances both the inward and outward current, as is evident in the $G/G_{max}$ vs. voltage relation.

## Polycystin-2 functions as a channel in primary cilia, but not in the plasma membrane

The above results confirm the location of functional ciliary polycystin-2 channels. Since the single channel recordings are made from the tips of cilia, polycystin-2 is present on the cilia membrane itself, not just the cilium/plasma membrane junction. However, native polycystin-2 channel are reportedly constitutively active in the plasma membrane of immortalized cell lines derived from kidney epithelial cells (mIMCD-3 and Madin-Darby canine kidney, MDCK, cells) (*Luo et al., 2003*). To test this possibility, we voltage clamped the plasma membrane of pIMCD cells harvested from *Arl13b-EGFP^tg:cPkd2* mice (*Figure 7A*; *Video 2*). Here, we typically observed an outwardly-rectifying $Na^+$-permeant current, an inwardly-rectifying $K^+$ current, and an apparent voltage-gated $Ca^{2+}$ current (*Figure 7B*). However, when polycystin-2 was conditionally reduced, there was no difference in plasma membrane currents densities (*Figure 7C*). Polycystin-2 function has been implicated in calcium transients originating in the endoplasmic reticulum (ER), plasma membrane, and cilium (*Ma et al., 2005*; *Nauli et al., 2003*; *Qian et al., 2003*). However, the plasma membrane current-voltage relationship, inactivation kinetics and pharmacology are typical of L-type calcium currents and did not change when polycystin-2 was reduced in these cells (*Figure 7D,E*). Thus, our findings suggest that polycystin-2 does not constitute a significant portion of the plasma membrane current found in primary collecting duct epithelial cells and does not alter the native voltage-dependent calcium current.

**Table 2.** Transmembrane reversal potentials ($E_{rev}$) measured from pIMCD and HEK-293 PKD2-GFP cilia.

| External solution | pIMCD, whole-cilium | | HEK-293 PKD2-GFP, whole-cilium | |
|---|---|---|---|---|
| | $\Delta E_{rev}$ (mean ± S.D.) | $P_x/P_{Na}$ (mean ± S.D.) | $\Delta E_{rev}$ (mean ± S.D.) | $P_x/P_{Na}$ (mean ± S.D.) |
| NaCl | 0 | 1 | 0 | 1 |
| NaMES | −2 ± 3 mV | 0.96 | Not tested | - |
| KCl | 26 ± 3 mV | 2.4 ± 0.3 | 23 ± 4 mV | 2.4 ± 0.4 |
| CaCl$_2$ | −49 ± 4 mV | 0.06 ± 0.04 | −48 ± 3 mV | 0.09 ± 0.04 |
| NMDG | −57 ± 4 mV | 0.04 ± 0.02 | −58 ± 4 mV | 0.04 ± 0.02 |

DOI: https://doi.org/10.7554/eLife.33183.012

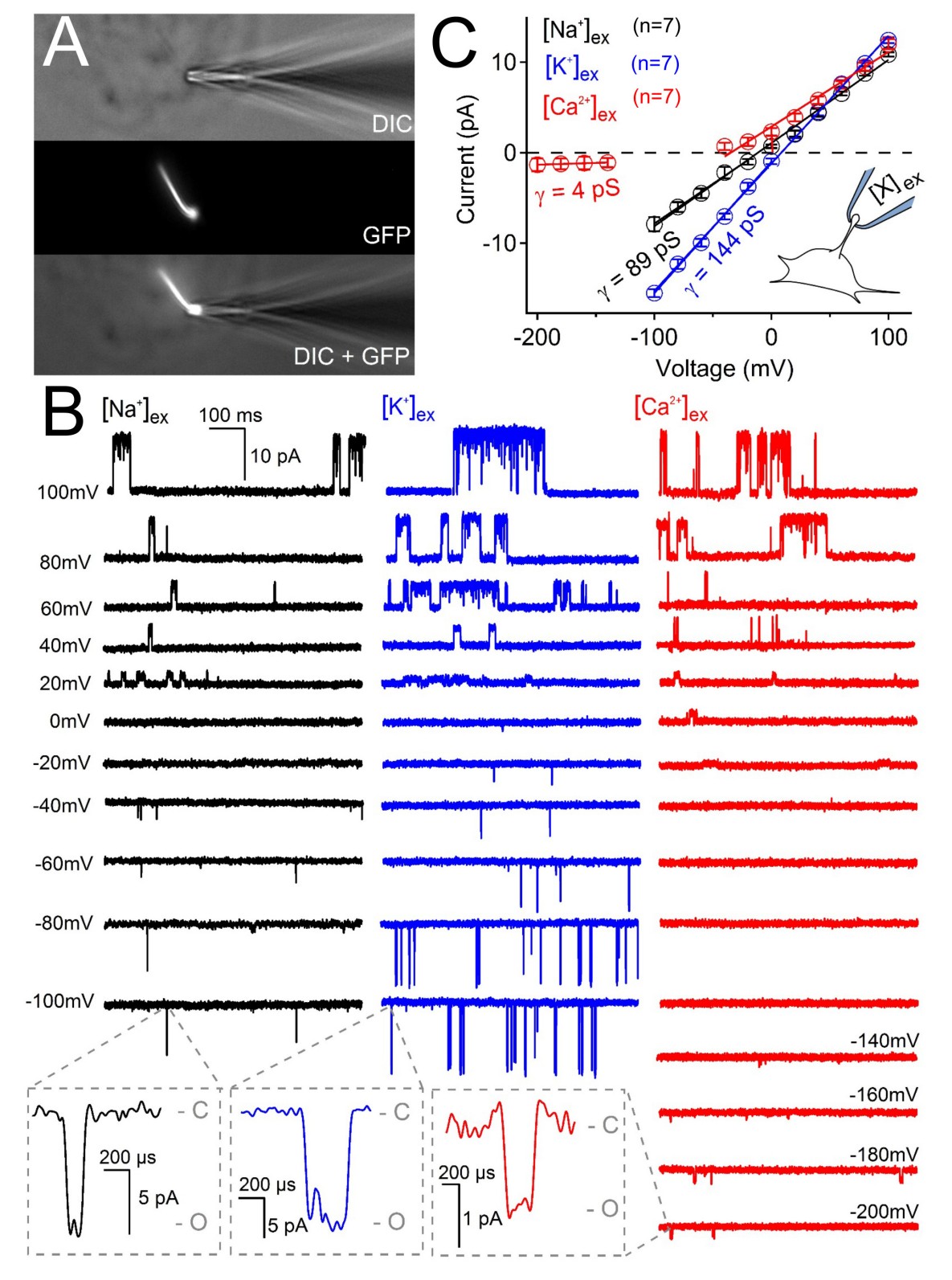

**Figure 5.** Ciliary polycystin-2 single channel currents conducted by sodium, potassium and calcium ions. (A) Image of a cilia patched without breaking into the cilioplasm ('on-cilium' configuration). (B) Exemplar currents recorded with the indicated cation (110 mM) in the patch electrode. Expanded time scales in the grey boxes show that inward single channel currents are brief, often opening (O) and closing (C) within 1 ms. (C) Average single channel current amplitudes. Conductance (γ) estimated by fitting the average single channel currents to a linear equation. Note that the inward single channel

*Figure 5 continued on next page*

*Figure 5 continued*

events are small (~0.8 pA at −200 mV) when $Ca^{2+}$ is used as the charge carrier in the pipette (inset, patch diagram); note outward currents are much larger. Outward conductances of 90 pS, 99 pS and 117 pS, and inward conductances of 4 pS, 89 pS, and 144 pS were measured when the pipette contained $Ca^{2+}$, $Na^+$ and $K^+$, respectively. *Inset*, a cartoon of the 'on-cilium' patch configuration where cations within the patch electrode ($[X^+]$) are exclusively capable of conducting inward currents.

DOI: https://doi.org/10.7554/eLife.33183.013

The following figure supplements are available for figure 5:

**Figure supplement 1.** The pIMCD ciliary polycystin-2 cilia membrane is highly permeable to $K^+$ and $Na^+$.

DOI: https://doi.org/10.7554/eLife.33183.014

**Figure supplement 2.** pIMCD single channel events are dependent on polycystin-2 expression.

DOI: https://doi.org/10.7554/eLife.33183.015

## Polycystin-2 is a ciliary channel but is not constitutively active on the plasma membrane in HEK-293 cells

Previously, our attempts to record heterologously-expressed polycystin-2 currents from the plasma membrane using transient transfection from multiple cell lines were unsuccessful. Thus, we generated two HEK-293 stable cell lines which overexpressed either human *PKD2* with the C-terminus (*PKD2-GFP*) or N-terminus tagged with GFP (*GFP-PKD2*). In fixed preparations and live cells viewed with confocal and standard fluorescence microscopy, GFP-polycystin-2 was intracellular, while polycystin-2-GFP also localized to primary cilia (*Figure 7—figure supplement 1A*; *Figure 7—figure supplement 2A,B*; *Video 3*). The ciliary localization of polycystin-2-GFP in HEK-293 cells was confirmed in super-resolution images in which GFP co-localized with the known ciliary proteins, acetylated tubulin, and adenylyl cyclase (AC3) (*Video 4*; *Video 5*, respectively). However, we could not rule out polycystin-2 functioning in the plasma membrane of *PKD2-GFP* cells by fluorescence alone. To address this, we voltage clamped the HEK-293 *PKD2-GFP* whole-cell ion currents and found no difference when compared to parental HEK-293 cells (*Figure 7—figure supplement 1D*, surface area of cilia <2% of total plasma membrane). Since native ciliary pIMCD polycystin-2 channels preferentially conduct $K^+$, we compared the plasma membrane potassium current using steady-state voltage protocols. The resulting current-voltage relationship and block by 4-AP (4-aminopyridine, a $K^+$ channel antagonist) suggests that the HEK-293 $K^+$ current is conducted by native voltage-gated potassium channels (*Figure 7—figure supplement 1C*), commonly reported in these cells (*Foley and Boccuzzi, 2010*; *Sands and Layton, 2014*; *Spandidos et al., 2008*; *Wilkes et al., 2017*). Importantly, the kinetics and magnitude of the plasma membrane potassium current was not altered when *PKD2-GFP* was stably overexpressed (*Figure 7—figure supplement 1D*). Thus, overexpressed *PKD2-GFP* by itself does not appear to form a constitutively active or voltage-gated channel in the nonciliary plasma membrane.

In contrast to the whole-cell recordings above, when we patch clamped the ciliary membrane of the *PKD2-GFP* HEK-293 cells, a large outwardly rectifying single channel conductance ($\gamma = 105 \pm 4$ pS) was observed (*Figure 7—figure supplement 2C–E*). The *PKD2-GFP* HEK-293 ciliary inward conductance ($\gamma = 90 \pm 3$ pS) was similar to the native polycystin-2 channels found in pIMCD cells ($\gamma = 89 \pm 4$ pS) when sodium was used as a charge carrier. In the whole-cilium configuration, the cilia membrane primarily conducted $K^+$ and $Na^+$ with little permeation by $Ca^{2+}$ (*Figure 7—figure supplement 3A,B*). To determine the identity of the outward rectifying current, we tested two siRNAs targeted to the overexpressed human *PKD2* and observed a 29% and 32% decrease in outward current compared to scrambled controls (*Figure 7—figure supplement 3D*, *Figure 7—figure supplement 3D—source data 1*-ciliary current amplitudes of cells treated with *PKD2*-mRNA-targeted siRNAs). Endogenous human polycystin-2 channels may be present in the cilia of HEK-293 cells and may contribute to some of the measured current. Future work genetically removing endogenous *PKD2* expression in HEK-293 cells will be necessary to study polycystin-2 channels in isolation and the impact of ADPKD-causing variants without the possibility of contaminating endogenous channels. To determine the localization of the channels mediating this current, we removed the cilia from the cell body by obtaining excised whole-cilium configuration recordings (*Figure 7—figure supplement 3C*). Excised patches from *PKD2-GFP* overexpressing HEK-293 cells retained an average of $44 \pm 21\%$ of the whole-cilium current. This percentage is highly variable since the amount of membrane

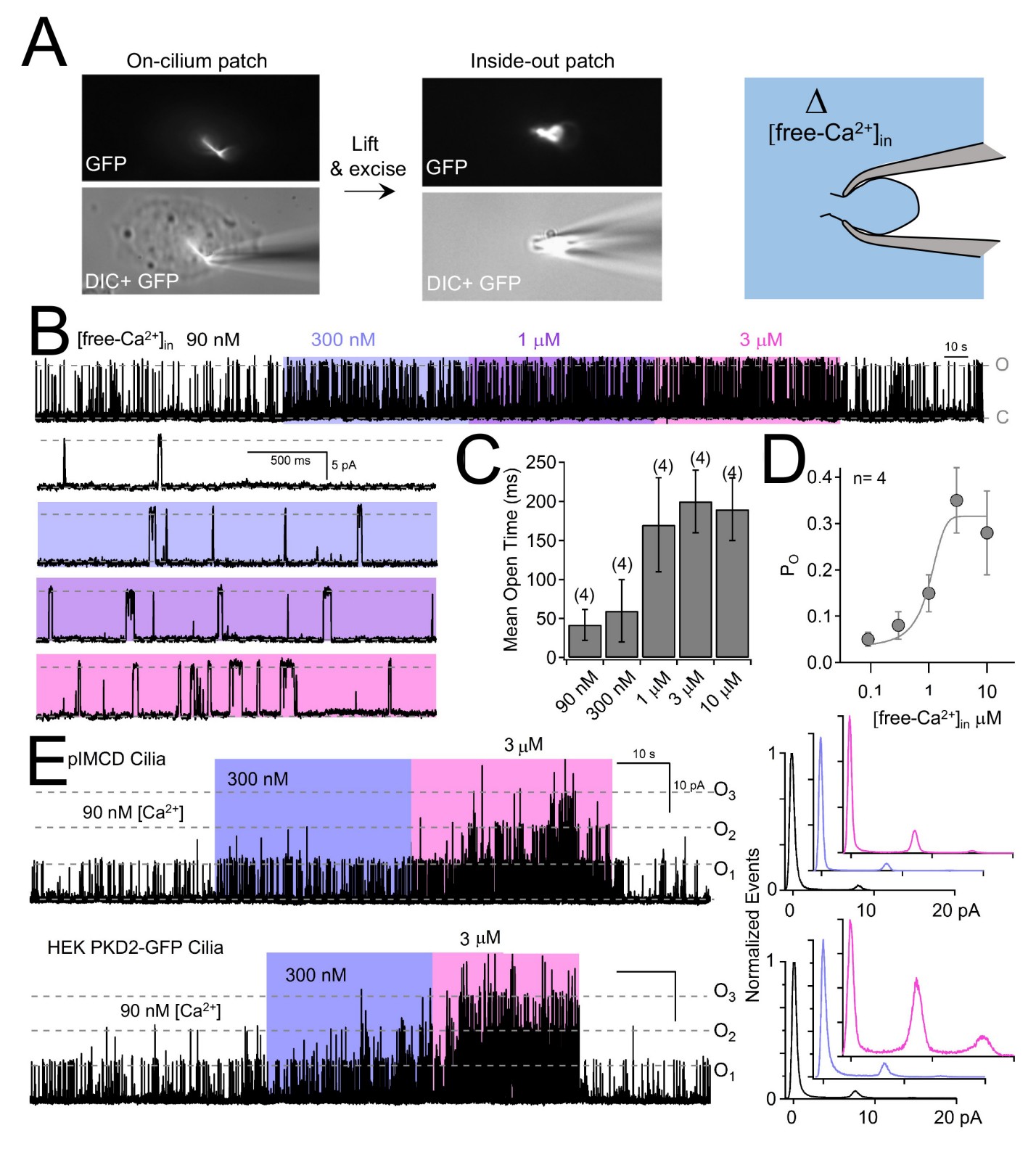

**Figure 6.** Internal [free-Ca$^{2+}$] potentiates polycystin-2 channels. (**A**) Images recorded while establishing an inside-out cilium patch. *Left*, a high-resistance seal is formed on the cilium; *Middle*, the electrode is then lifted, ripping the cilium from the cell body (see Materials and methods); *Right*, cartoon depicting the inside of the cilium exposed to bath saline (blue) in which [Ca$^{2+}$] can be adjusted. (**B**) pIMCD polycystin-2 single channel events recorded in the inside-out configuration. The membrane potential was held at +100 mV in symmetrical [Na$^+$] while the internal [Ca$^{2+}$] was altered for

*Figure 6 continued on next page*

*Figure 6 continued*

40–60 s intervals. (C) The average open time duration relative to internal [Ca$^{2+}$] (mean ± S.D.). (D) Average open probability as a function of internal [Ca$^{2+}$], fit to the Hill equation (described in Materials and methods, mean ± S.D.). (E) *Left*, exemplar inside-out cilium patch records from pIMCD cilia and HEK-293 cilia with heterologously expressed polycystin-2 channels. *Right*, current histograms capturing multiple open channel events under high internal [Ca$^{2+}$] conditions. Currents were normalized to the closed (0 pA) state amplitude for each internal [Ca$^{2+}$].
DOI: https://doi.org/10.7554/eLife.33183.016
The following figure supplement is available for figure 6:

**Figure supplement 1.** Internal calcium hyperpolarizes polycystin-2's voltage dependence.
DOI: https://doi.org/10.7554/eLife.33183.017

preserved after excision varies between patches. Thus, it is unclear if the missing 56% of the ciliary current originated from the disconnected cilia membrane or near the cilia-plasma membrane junction. As found for the polycystin-2 channels in the pIMCD cilia, increasing internal calcium stimulated multiple open polycystin-2 channels in the inside-out patch configuration of HEK-293 cilia (*Figure 6E*). Thus, ciliary polycystin-2 currents on *PKD2-GFP* overexpressing HEK-293 cells reproduces the ion selectivity and internal calcium sensitization as found in the native polycystin-2 channels of pIMCD cilia.

## Discussion

### *Arl13b-EGFP$^{tg}$:cPkd1* and *Arl13b-EGFP$^{tg}$:cPkd2* as new mouse models for cilia visualization during ADPKD progression

We confirmed cyst progression upon reduction of *Pkd1* or *Pkd2* in adult renal collecting duct epithelia. Because the mice also expressed the *Arl13b-EGFP$^{tg}$* transgene, we could compare the effects of reduced polycystins on cilia formation. Conditional ablation of either gene did not block the formation of cilia in adult collecting ducts, consistent with findings in the embryonic node, where constitutive global repression of polycystin-2 had no effect on cilia number (*Field et al., 2011*). Unexpectedly, we observed elongated and twisted cilia, a feature that became more pronounced as the cystic phenotype progressed. These data support the hypothesis that ciliary polycystin-1 and polycystin-2 are essential to the maintenance of normal renal tubular cell differentiation. It is important to point out that the observed altered ciliary morphology and cyst formation are not necessarily causally related. Currently, we do not understand how loss of polycystins affect cilia morphology, but hypothesize that they alter ciliary transport or modification of proteins shuttling through or sequestered within cilia.

### Polycystin-2 is primarily a monovalent channel in the cilium

A commonly held hypothesis is that polycystin-1 and polycystin-2 form a calcium-permeant channel directly involved in aberrant cytoplasmic calcium signaling (*Pei, 2001*). Previous work measuring reconstituted polycystin-2 channels from heterologous and native sources report conflicting voltage sensitivity and ion selectivity (*Pablo et al., 2017*). Here, by directly measuring channels in primary cilia, we have shown that polycystin-2 is an essential subunit for the outwardly rectifying current. In primary cilia of native kidney tubular epithelial cells, polycystin-2 current is relatively selective for monovalent cations ($P_x/P_{Na}$ = 2.4 and 1, for K$^+$ and Na$^+$, respectively), with comparatively little calcium permeation (Ca$^{2+}$ ~ NMDG).

The cilium, like the dendrite of neurons, is a quasi-distinct compartment, enforced by >100 MΩ resistance between the volumes, >100 fold differences in surface areas (~10 pF vs ~100 fF), distinct proteins with distinct ion binding characteristics, and restricted diffusion. Nonetheless, since there is no membrane separation between compartments, it is important to measure these compartments separately, as we previously reported (*DeCaen et al., 2013*). The pIMCD and HEK whole-cell currents capture a distinct set of ion channels from those measured in the whole-cilium configuration. As confirmed by excised whole-cilium recordings, currents from the whole-cell and whole-cilium can be reliably separated. Currents from these membranes can also be distinguished by changing external cations (compare *Figure 7A–C*, *Figure 7—figure supplement 1A–B* with *Figure 4B*). The pIMCD whole-cell cationic currents (*Figure 7B*) consist of an inwardly rectifying potassium current (Kir), a

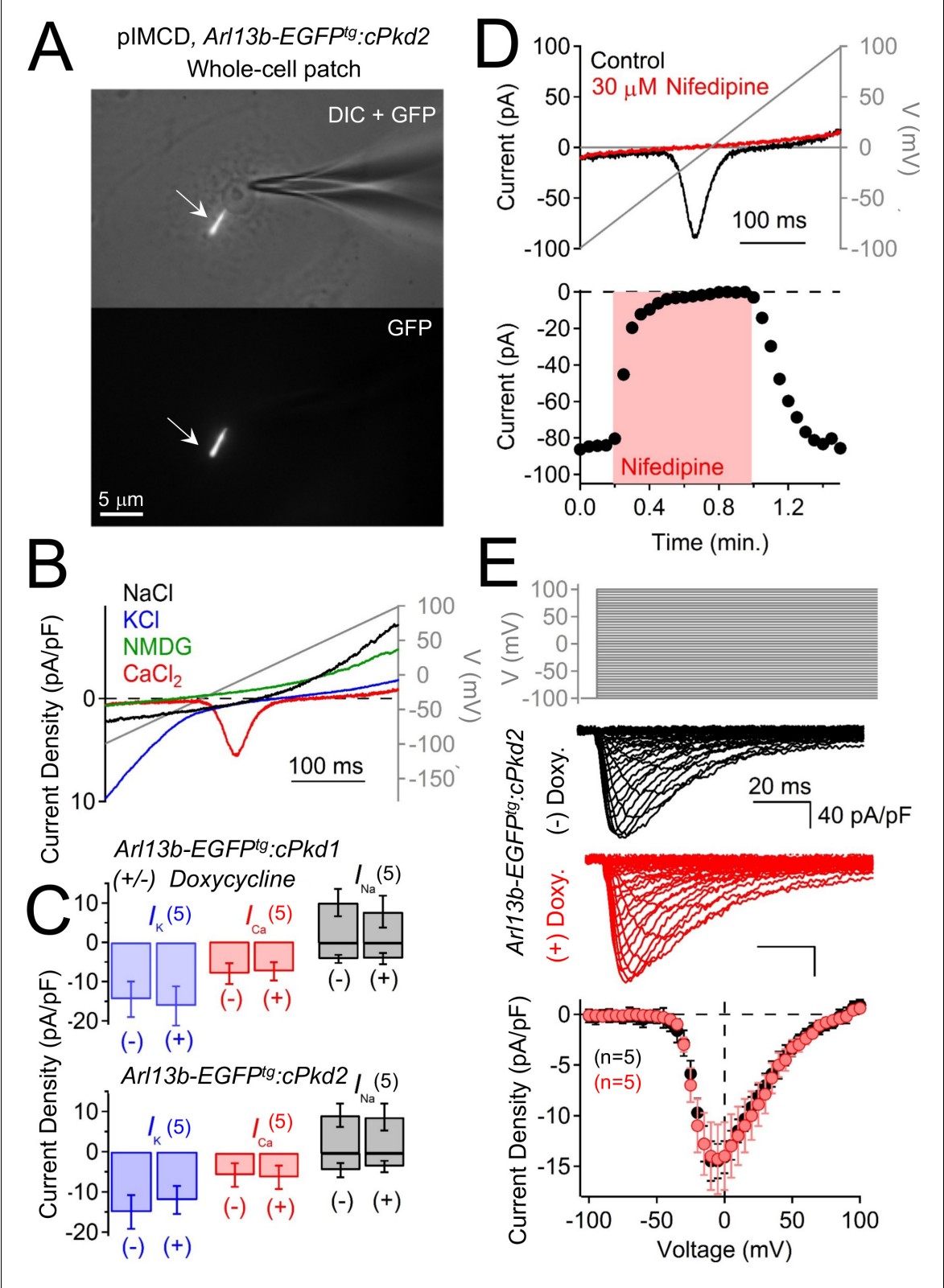

**Figure 7.** Polycystin-2 does not generate a significant current in the plasma membrane of pIMCD cells. (A) Images of a ciliated pIMCD epithelial cell patch-clamped on *the plasma membrane* (whole-cell mode). Scale bar = 5 μM. (B) Example plasma membrane ionic currents activated by a voltage ramp (grey line) while changing the extracellular saline conditions. (C) Average plasma membrane current density of *Arl13b-EGFP^tg:cPkd2,* where animals treated with doxycycline was compared to untreated animals. Note that the average magnitudes of the plasma membrane currents were not

*Figure 7 continued on next page*

*Figure 7 continued*

significantly altered (mean ± S.E.M.). (**A**). (**D**) Pharmacological blockade of the voltage-gated calcium channel in the plasma membrane; three independent experiments. *Top,* exemplar calcium currents blocked by nifedipine. *Bottom,* time course of block and recovery of Ca$_V$ currents. (**E**) Conditional polycystin-2 knockout does not alter the steady state voltage-gated calcium currents measured from the plasma membrane. *Top,* voltage protocol used to activate the calcium currents. *Middle,* exemplar leak-subtracted voltage-gated calcium currents from doxycycline-treated (red) and – untreated (black) *Arl13b-EGFP$^{tg}$:cPkd2* animals. Resulting plasma membrane average Ca$_V$ densities compared from doxycycline-treated and control littermates.

DOI: https://doi.org/10.7554/eLife.33183.018

The following source data and figure supplements are available for figure 7:

**Figure supplement 1.** Characterization of plasma membrane currents measured from HEK-293 cells stably expressing polycystin-2-GFP.

DOI: https://doi.org/10.7554/eLife.33183.019

**Figure supplement 2.** Single channel events recorded from cilium of polycystin-2-GFP overexpressing HEK-293 cells.

DOI: https://doi.org/10.7554/eLife.33183.020

**Figure supplement 3.** Overexpressed polycystin-2 forms an ion channel in the HEK-293 cilium.

DOI: https://doi.org/10.7554/eLife.33183.021

**Figure supplement 3—source data 1.** Ciliary current amplitudes of cells treated with *PKD2*-mRNA-targeted siRNAs.

DOI: https://doi.org/10.7554/eLife.33183.022

voltage-gated calcium current (Ca$_v$), and an outwardly-rectifying sodium current. In contrast, the pIMCD whole-cilium current is a non-selective outwardly-rectifying current dependent on polycystin-2 expression (*Figure 4B*).

Kleene and Kleene identified polycystin-2 as a large conducting ion channel (outward γ = 96 pS) of mIMCD-3 cells (*Kleene and Kleene, 2017*). mIMCD-3 are immortalized epithelial cells derived from the terminal portion of the inner medullary collecting duct of SV40 transgenic mice and have similar morphology to the primary IMCD epithelial cells used in our study. We observed a similar outward conductance (γ = 90–117 pS) in the cilia of primary collecting duct epithelial cells directly harvested from adult mice. Like the results reported from the mIMCD-3 cell lines, unitary and whole-cilium currents from the pIMCD primary cells were reduced or absent when polycystin-2 was knocked down with siRNAs or conditionally knocked-out in the whole animal. Both studies note the sensitization of the current by μM [Ca$^{2+}$]. Based on these similarities, it is likely that we are describing the same ciliary polycystin-2 channel in these cell types. Both studies agree that the polycystin-2 cilia conductance is most selective for potassium but differ in estimates of sodium and calcium permeability (P$_X$/P$_K$ = 1: 0.14: 0.55 for K$^+$, Na$^+$ and Ca$^{2+}$, respectively) (*Kleene and Kleene, 2017*) compared to P$_X$/P$_K$ = 1: 0.4: 0.025 for K$^+$, Na$^+$ and Ca$^{2+}$, respectively, in this study. To compare readily with our previous work describing other ciliary polycystin channels (*Shen et al., 2016*; *DeCaen et al., 2016*; *DeCaen et al., 2013*), we report P$_X$/P$_{Na}$ = 2.4, 1, and 0.06 for K$^+$, Na$^+$, and Ca$^{2+}$, respectively.

The relative permeabilities of Na$^+$ and K$^+$ are relatively consistent between Kleene and Kleene, Pavel et al, and our results. The ~20 fold difference in P$_{Ca}$/P$_K$(0.55 vs 0.025) most probably arises from the linear extrapolation by Kleene et al to obtain the E$_{rev}$. Other differences may be relevant: we recorded from primary tubule cells (pIMCD), not from immortalized cells (mIMCD-3; near-triploid karyotype) (*Battini et al., 2008*). Second, external [Ca$^{2+}$] was chelated to <1 nM (and Mg$^{2+}$<10 μM) in our solutions in which inward monovalent currents were measured. Kleene and Kleene added 2 mM Mg$^{2+}$ and varying [Ca$^{2+}$] in both internal and external conditions (*Kleene and Kleene, 2017*). Finally, the method utilized by Kleene and Kleene

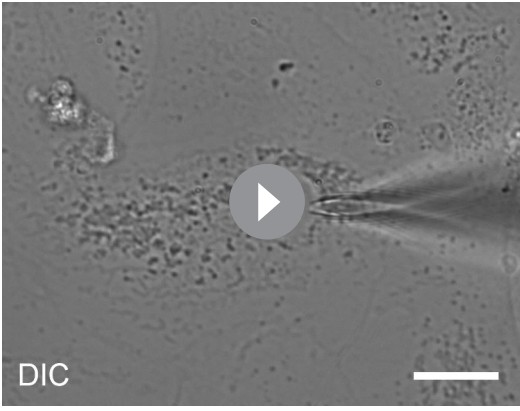

**Video 2.** Visualization of a plasma membrane patch from a ciliated pIMCD *Arl13b*-GFP cell. Same method as *Video 1*. The cell membrane is being sealed by the patch electrode (*right*) and the primary cilium (*left*) is not in contact with the electrode. Scale bar = 10 μm.

DOI: https://doi.org/10.7554/eLife.33183.023

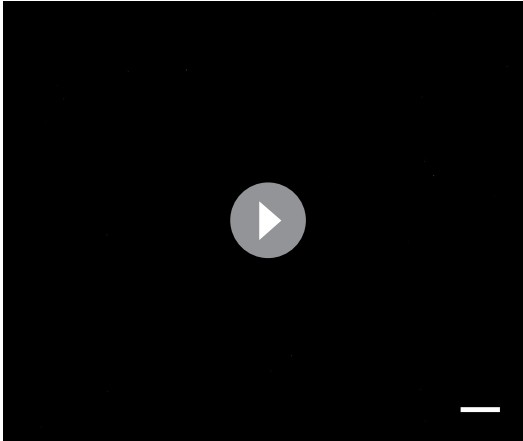

**Video 3.** A plasma membrane patch established on a polycystin-2-GFP HEK-293 cell. Here the whole-cell configuration is formed by the patch electrode (*left*) on the cell body and the primary cilium (*lower left*) is not in contact with the electrode. The electrode is moved along the y-axis while adjusting the focal plane to demonstrate that the cell membrane is sealed on the patch electrode, not the cilia membrane. Scale bar = 5 µm.
DOI: https://doi.org/10.7554/eLife.33183.024

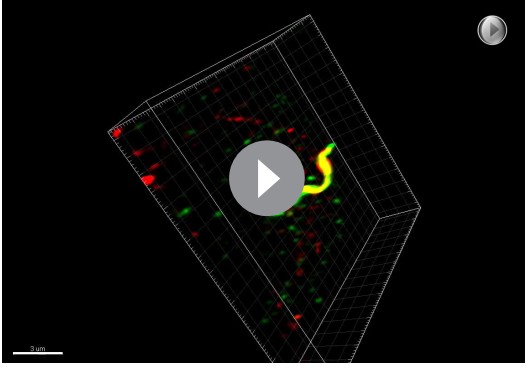

**Video 4.** Ciliary co-localization of polycystin-2-GFP with immunolabeled acetylated tubulin. Paraformaldehyde-fixed HEK-293 cells stably expressing polycystin-2-GFP (*green*) were immunolabeled with anti-acetylated tubulin antibody (*red*). The images were captured using Structured Illumination Microscopy (SIM; Nikon N-SIM scope) and 3D images rendered using Imaris software (Oxford Instruments). Scale bar = 3 µm.
DOI: https://doi.org/10.7554/eLife.33183.025

envelops the cilium and its base within the recording pipette, thus including some plasma membrane (*Kleene and Kleene, 2012*). Here, we patch only ciliary membrane, albeit with some of its base removed (see *Video 1*) (*DeCaen et al., 2013*). To help refine our measurement, we estimated the relative permeability based on the extrapolated $E_{rev}$ from our single channel currents (*Figure 5—figure supplement 2D*). The ion selectivity results from our single channel and whole-cilium measurements are consistent, which strengthens our conclusions regarding the rank order of cation selectivity; $K^+ > Na^+ >> Ca^{2+}$.

It is important to note that neither we nor the Kleene group found inward calcium-mediated single channel events from potentials ranging from 0 mV to −100 mV (personal communication with Steven Kleene, Univ. of Cincinnati). However, we were able to resolve unitary single channel events under non-physiological conditions in which the external calcium was high (110 mM) and the cilium's membrane potential was very hyperpolarized (more negative than −120 mV). Thus, there would be little calcium influx into cilia under physiological conditions since the resting cilia membrane potential is only −18 mV (*Delling et al., 2013*). Nonetheless, aberrant calcium signaling has been observed from cells expressing mutant polycystin-2 channels and interpreted as due to polycystin-2 function as an ER calcium-release channel (*Cai et al., 2004*; *Ćelić et al., 2012*), a function we have not investigated. Since $Ca^{2+}$ changes in the cytoplasm propagate into cilium (*Delling et al., 2016*), we cannot rule out the possibility that polycystin-2 in the ER alters $[Ca^{2+}]_{cilium}$. Also, although polycystin-2's $Ca^{2+}$ conductance is small, the cilium is a < 1 fL restricted space in which localized proteins might be influenced directly by occasional ciliary polycystin-2 channel $Ca^{2+}$ flux. However, it should not be overlooked that the major consequence of polycystin-2's selectivity is to depolarize the cilium and raise $[Na^+]_{cilium}$. If $Na^+/Ca^{2+}$ exchangers or $Na^+$-dependent kinases are found in cilia, polycystin-2 activity could underlie a slow, cumulative signal via calcium changes and/or kinase activity.

## Potential polycystin-2-specific function in kidney

Renal epithelial cilia are exposed to urine, which contains varying ion concentrations as a function of position in the nephron. Human and murine distal collecting duct epithelial cells are exposed to high external concentrations of potassium (90–260 mM) and sodium (53–176 mM) ions, which contributes to the hyperosmolarity of urine (390–650 mOsm being considered normal, but can vary beyond this range depending on hydration state) (*Callís et al., 1999*; *Sands and Layton, 2014*). Ciliary influx through polycystin-2, driven by these extreme extracellular concentrations of $Na^+$ and $K^+$ ions, may

depolarize the plasma membrane sufficiently to activate voltage-gated calcium channels present in the plasma membrane. A recent computational study finds that opening of single ciliary polycystin 2-L1 channel (~150 pS) is sufficient to trigger action potentials in the soma of cerebrospinal fluid-contacting neurons at standard concentrations in blood plasma (*Orts-Del'Immagine et al., 2016*). Future electrophysiological studies using current clamp will determine whether the activation of ion channels in the ciliary membrane is sufficient to depolarize the plasma membrane of pIMCD cells, if there is any polycystin-1-dependent polycystin-2 function that has different ion selectivity or permeability compared to polycystin-1-independent function, or if there are more direct consequences of polycystin-2 expression for other ciliary compartment proteins.

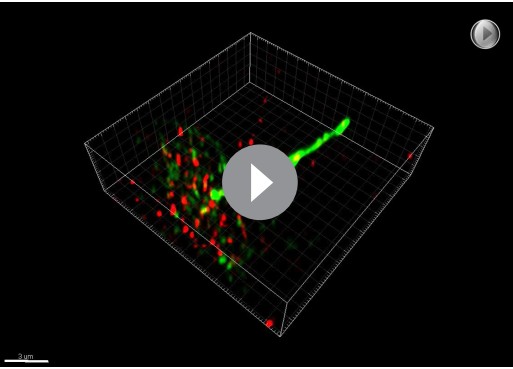

**Video 5.** Cilia co-localization of polycystin-2-GFP with immunolabeled adenylyl cyclase 3. Paraformaldehyde-fixed HEK-293 cells stably expressing polycystin-2-GFP (*green*) were immunolabeled with anti-adenylyl cyclase three antibody (*red*). The images were recorded using a Nikon N-SIM scope and 3D images rendered using Imaris software (Oxford Instruments). Scale bar = 3 μm.
DOI: https://doi.org/10.7554/eLife.33183.026

## Ciliary polycystin-2 channels are activated by internal calcium – relevance to kidney

We have demonstrated that extraciliary $Ca^{2+}$ can block monovalent conductance through polycystin-2, a common phenomenon in selective and non-selective cation channels, and responsible for the anomalous mole fraction effect (*Eisenman et al., 1986*; *Friel and Tsien, 1989*; *Sauer et al., 2013*). This effect is likely due to a relatively higher affinity for $Ca^{2+}$ ions in the pore ($IC_{50}$ = 17 mM), thus blocking the channel to the inward passage of $Na^+$ and $K^+$ ions. Recently, polycystin-2 structures were captured in the 'single' and 'multi-ion mode' states; 20 mM $Ca^{2+}$ and 150 mM $Na^+$ were present during protein purification. $Ca^{2+}$ bound at the entrance of the selectivity filter suggests either a low $Ca^{2+}$ conducting state or a blocked state, in which $Ca^{2+}$ ions prevent $Na^+$ ion permeation in the multi-ion mode (*Grieben et al., 2017*; *Wilkes et al., 2017*). These studies provide a structural context for the anomalous mole fraction effects we observe in polycystin-2 currents from the pIMCD cilia and those reported by other groups measuring polycystin-2 channels from other preparations (*Arif Pavel et al., 2016*; *Cai et al., 2004*; *Koulen et al., 2002*; *Vassilev et al., 2001*). What, if any, effect might anomalous mole fraction effects have on the ciliary polycystin-2 channel? In contrast to the physiological, typically tightly-controlled interstitial $[Ca^{2+}]$ (~1.8 mM), urinary $[Ca^{2+}]$ in humans and mice is highly variable (5–20 mM) during normal diurnal activity (*Foley and Boccuzzi, 2010*). Urinary $Ca^{2+}$ may have physiologically-relevant effects in dynamically limiting the polycystin-2 monovalent current through the cilium. When calcium-wasting occurs (~15 mM tubular $[Ca^{2+}]$), more than half of $Na^+$ and $K^+$ influx through polycystin-2 would be antagonized (see *Figure 5—figure supplement 1*). Thus, ciliary polycystin-2 would be most active during low $[Ca^{2+}]$ in the tubule urine. On the other side of the cilia membrane, shifting the internal $[Ca^{2+}]$ from resting levels (600 nM) to 30 μM activates the polycystin-2 current by increasing the open probability and the total inward conductance ~10 fold. Internal $Ca^{2+}$-dependent potentiation has been reported in polycystin-2 channels reconstituted from the ER into bilayers, but these channels inactivate at $[free-Ca^{2+}]_{in}$ concentrations > 1 μM unless mutated at C-terminal phosphorylation sites (*Cai et al., 2004*). Previous measurements of resting $[free-Ca^{2+}]_{in}$ from RPE cilia (580 nM) (*Delling et al., 2013*) and from mouse embryonic node cilia (305 nM) (*Delling et al., 2016*) are 3–6 times higher than in the cell body. Primary cilia $[free-Ca^{2+}]_{in}$ increases to levels greater than 1 μM when mIMCD-3 cells (and other cell types) are exposed to flow or shear stress (*Delling et al., 2013*; *Su et al., 2013*). Recent work has demonstrated that primary cilia are not $Ca^{2+}$-responsive mechanosensors themselves; rather, mechanically-induced calcium waves are initiated from other locations to raise ciliary $[Ca^{2+}]$ (*Delling et al., 2016*). Thus, increasing cytoplasmic $[free-Ca^{2+}]$ by mechanical or other stimuli, may increase cilioplasmic calcium and potentiate ciliary polycystin-2 channel activity. Consistent with the cilia channel recordings made by Kleene and Kleene (*Kleene and Kleene, 2017*), a 50x increase in the cilioplasmic calcium reduces the voltage

threshold required to activate polycystin-2 from our pIMCD cilia. If the cilia membrane potential of collecting duct cells is as depolarized as the cilia of RPE cells ($-18$ mV), then we can expect an ~10 x increasing polycystin-2 opening when cytoplasmic calcium waves reach the cilioplasm.

## Polycystin-2 and polycystin 2-L1, independently form ion channels in primary cilia in disparate tissues

Previously we characterized the ciliary current from retina pigmented epithelial cells (RPE) and mouse embryonic fibroblasts (MEF) and demonstrated that they require the polycystin family proteins, polycystin 1-L1 and polycystin 2-L1, based on attenuation of the cilia current by siRNA and genetic ablation of these two genes (*DeCaen et al., 2013*). Using similar methodology, including cilia electrophysiology from primary collecting duct cells, we have determined that polycystin-2 is at least a component of $I_{cilia}$ from these cells. Thus, polycystin-2 and polycystin 2-L1 may be ciliary ion channels inhabiting distinct cellular tissues. Although the single channel conductance and sensitization of the RPE and MEF *Pkd1-L1/2-L1*-encoded cilia channel (Inward $\gamma_{Na}$ = 80 ± 3 pS) (*DeCaen et al., 2013*) is similar to the pIMCD cilia polycystin-2 channel (Inward $\gamma_{Na}$ = 89 ± 4 pS), their $Ca^{2+}$ selectivity is distinct. Also, we showed here that the ciliary pIMCD polycystin-2 channel is blocked by external $Ca^{2+}$ but is sensitized by high (EC$_{50}$ = 1.2 μM) internal $Ca^{2+}$,~10 times the typical resting cytoplasmic concentration. Heterologous polycystin 2-L1 channels are also sensitized by increases in $[Ca^{2+}]_{in}$ (although the sensitivity range has not been determined) based on cytoplasmic $Ca^{2+}$ uncaging studies and expected $Ca^{2+}$ accumulation in whole cell experiments (*DeCaen et al., 2016*). However, a key difference between pIMCD (polycystin-2) and RPE (polycystin 2-L1) ciliary channels are that polycystin 2-L1 channels preferentially conduct $Ca^{2+}$ ($P_{Ca}/_{Na}$ = 6–19) (*DeCaen et al., 2013*; *DeCaen et al., 2016*) over monovalent ions. Mutagenesis studies of heterologous polycystin 2-L1 channels has demonstrated that $Ca^{2+}$ permeation is at least partly due to an additional glutamate residue (D525) on the external side of the selectivity filter, not present in polycystin-2 (*DeCaen et al., 2016*). The physiological implications for the differential cilia expression of polycystins and attendant differences in ion selectivity is not known.

## Polycystin-2 structural considerations

Soon after these studies, polycystin-2 core structures were solved using single-particle electron cryo-microscopy in which the polycystin-2 channels formed a homotetrameric structure (*Grieben et al., 2017*; *Shen et al., 2016*), independent of coiled-coil domains (originally posited to form an interaction motif with polycystin-1 subunits [*Qian et al., 1997*]). As reported (*Shen et al., 2016*), replacement of the polycystin 2-L1 filter with that of polycystin-2 conferred monovalent selectivity to the otherwise $Ca^{2+}$-permeant polycystin 2-L1 channel. However, it is important to note that the native ciliary polycystin-2 channel's ion selectivity, as described here, is not completely recapitulated in the polycystin-2 filter chimera, where single channel $Ca^{2+}$ conductance was~27 fold smaller than $K^+$ ($\gamma_{Ca}$ = 8 ± 2 pS, twice as large as the native cilia polycystin-2 channels, $\gamma_{Ca}$ = 4 ± 1 pS). Nonetheless, the native polycystin-2 cilia channel and the polycystin-2 filter chimera share similar ion selectivity, where $K^+$ is favored over $Na^+$ and $Ca^{2+}$ as reflected in the magnitudes of single channel conductance (pIMCD cilia $\gamma_K$ = 144 ± 6 pS, $\gamma_{Na}$ = 89 ± 4 pS; polycystin-2 chimera $\gamma_K$ = 218 ± 3 pS, $\gamma_{Na}$ = 139 ± 3 pS) and relative permeability ($P_x/P_{Na}$ pIMCD cilia = 2.4: 1: 0.06 and polycystin-2 chimera = 2.2: 1: 0.5 for $K^+$, $Na^+$ and $Ca^{2+}$ respectively).

## Loss of polycystin-1 does not alter polycystin-2 ciliary trafficking or polycystin-2 mediated ciliary currents

Based on the two-hit hypothesis of ADPKD, inherited haploinsufficiency of either polycystin-1 and polycystin-2 and a second acquired somatic mutation is required for disease progression. It was reported that interaction between polycystin-1 and polycystin-2 is required for cell plasma membrane trafficking of the complex (*Chapin et al., 2010*; *Gainullin et al., 2015*; *Hanaoka et al., 2000*), but their interdependence for ciliary localization is controversial. In some studies, polycystin-2 is absent from primary cilia without polycystin-1 or *vice versa* (*Gainullin et al., 2015*; *Kim et al., 2014*), while others showed polycystin-2 traffic to cilia is independent of polycystin-1 (*Cai et al., 2014*; *Geng et al., 2006*; *Hoffmeister et al., 2011*). Our data support the view that polycystin-2 can traffic to primary cilia of pIMCD cells in the absence of polycystin-1. The differences with some of the

previous reports may due to the different experimental systems. We employed isolated pIMCD cells with few or no passages, while others used cell lines (mIMCD3, HEK-293, LLC-PK1, or Renal Cortical Tubule Epithelial; RCTE) or cells overexpressing proteins. Another difference is that our mice express the cilia marker *Arl13b-EGFP*, which might affect ciliary trafficking. There are no other studies reporting this mouse model to date.

Based on biochemical data and initial whole-cell electrophysiology, coexpression of polycystin-1 and polycystin-2 were reported to be necessary and sufficient to form heteromeric $Ca^{2+}$-permeant channels in the cell soma plasma membrane (*Hanaoka et al., 2000*). However, we have noted a dearth of recordings in the literature, and of the few published, there are many inconsistencies. Based on the data here, and the selectivity measured from the polycystin-2 gating mutant (F604P) (*Arif Pavel et al., 2016*), polycystin-2 is primarily a monovalent channel with selectivity $K^+ >$ $Na^+$, whose current is blocked by extracellular $Ca^{2+}$. It resembles many TRP channels in being slightly outwardly rectifying under physiological conditions, with a large single channel conductance (~100 pS). Polycystin-1 and polycystin-2 were believed to form a channel by association with their C-terminal coiled-coil domain (*Qian et al., 1997*). In contrast, recent structures of polycystin-2 homo-tetramers channels suggest that it can form a pore in the absence of the polycystin-2-coiled-coil domain and without polycystin-1 subunits (*Grieben et al., 2017*; *Shen et al., 2016*; *Wilkes et al., 2017*). Furthermore, removal of the conserved coiled-coil does not alter homomeric functional assembly of the related polycystin 2-L1 channel (*DeCaen et al., 2016*; *Li et al., 2002*).

Our genomic PCR data (*Figure 1—figure supplement 2A and E*) cannot rule out the possibility that not all pIMCD cells completely lacked polycystin-1, although the more functionally-relevant patch clamp results show no current differences (*Figure 4*) between doxycycline-treated and untreated groups. Our conditional knockout mice are inducible knockouts of *Pkd1* or *Pkd2* in renal tubular epithelial cells in the Pax8rtTA; TetO-cre system, but not all tubule cells are knock-out cells (e.g., the S3 straight segment, where Cre activity was largely absent [*Ma et al., 2013*]). Non-tubule cells like fibroblasts, interstitial cells, and endothelial cells may also affect the purity of the cells we isolated. Our conclusions are primarily based on patch clamp electrophysiology, which is more sensitive and functionally relevant than western blots, tagging, or immunohistochemistry. We ablated polycystin-1 and still observe the same magnitude of polycystin-2 currents from the discrete primary cilium (*Figure 4B*). If polycystin-1 is necessary – either as a polycystin-2 chaperone or part of the functional channel complex – its impact on polycystin-2 ion channels should have been detected with electrophysiological recordings. It is important to note that our doxycycline-dependent ablation of polycystin-1 (or polycystin-2) is sufficient to produce the cystic kidney phenotype in these mice. Thus, reduced levels, whether complete or not, are sufficient for the observed pathophysiological changes.

As a final caveat, our results cannot exclude the possibility that polycystin-1 may associate with polycystin-2 in the cilium or perhaps in the membrane of the endoplasmic reticulum or Golgi. It is also possible that polycystin-1 may still modulate ciliary currents, perhaps indirectly as a receptor for extracellular ligands, or through direct association. However, our findings demonstrate that polycystin-1 is not essential for basal activity of polycystin-2 in primary cilia of pIMCD cells. Further work will likely refine the mechanism of the two-hit model of ADPKD progression.

## Polycystin-2 is not a constitutively active ion channel in the plasma membrane

In previous work, we established that polycystin 2-L1 can form a constitutively active ion channel in the plasma membrane and in primary cilia (*DeCaen et al., 2013*, *DeCaen et al., 2016*). We also reported that polycystin-2 did not appear to function on the plasma membrane, where HEK-293 and CHO cells transiently overexpressing polycystin-2, with or without polycystin-1, have the same level of plasma membrane cation currents observed in untransfected cells (*DeCaen et al., 2013*; *Shen et al., 2016*). In this manuscript, we present several lines of evidence suggesting that polycystin-2, unlike polycystin 2-L1, does not constitutively function in the plasma membrane in kidney epithelial cells. First, conditional knockout of polycystin-2 does not alter the major cation currents found in the plasma membrane of primary collecting duct epithelial cells. Second, we did not observe differences in plasma membrane current measured from HEK-293 cells stably-expressing *PKD2-GFP* in parental cells. While it is possible that polycystin-2 in the plasma membrane could be stimulated by a ligand such as Wnt3a, Wnt9b, and triptolide, we have not been able to reproduce activation of

heterologous polycystin-2 with these reagents (*Kim et al., 2016*; *Leuenroth et al., 2007*; *Ma et al., 2005*). Functional polycystin-2, heterologously expressed in the plasma membrane of *Xenopus laevis* oocytes, required an F604P mutation near its intracellular gate (*Arif Pavel et al., 2016*), similar to mutations required for TRPML plasma membrane function (*Grimm et al., 2012*; *Xu et al., 2007b*). The polycystin-2 F604P current was selective for potassium and sodium, but blocked by extracellular calcium and magnesium. Our attempts to record overexpressed polycystin-2 F604P membrane currents in mammalian cells have been unsuccessful (*Shen et al., 2016*), but such expression in *Xenopus* oocytes has been reproduced (unpublished data, courtesy of Michael Sanguinetti, Univ. Utah). What is clear is that *wt* polycystin-2 has no measurable constitutive activity in the plasma membrane in either mammalian or *Xenopus* expression systems.

## Polycystin-2-GFP cilia recording for detection of heterologous expression

Other groups have observed epitope-labeled polycystin-1 and polycystin-2 in HEK-293 cell primary cilia (*Gerdes et al., 2007*; *Lauth et al., 2007*). C-terminal GFP-tagged polycystin-2 enriches in primary cilium when stably expressed in HEK-293 cells. We observed that N-terminal GFP-tagged polycystin-2 fails to localize to the cilium, suggesting that the sensor in this position interferes with trafficking. It is possible that the N-terminal GFP tag may interfere with the amino-terminal cilia-localization sequence ($R_6VXP$)(*Geng et al., 2006*) and likewise, the C-terminal GFP tag may interfere with the ER retention sequence found in the C-terminus (*Cai et al., 1999*). However, since we have demonstrated that native (untagged) polycystin-2 is functionally expressed in cilia of collecting duct epithelia, the C-terminally-tagged polycystin-2 over-expressed in HEK-293 cilia appears to properly localize. Like many overexpressed ion channels, we observed a high amount of GFP fluorescence from N- and C-terminally-tagged polycystin 2-L1 within intracellular compartments. Polycystin-2 in the ER has been shown to be sensitized by cytoplasmic calcium, triggering $Ca^{2+}$-induced $Ca^{2+}$ release, possibly through direct interaction with the IP3R channel (*Koulen et al., 2002*; *Vassilev et al., 2001*). However, we did not examine polycystin-2 function in the ER or how cilia polycystin-2 may alter intracellular store calcium release. Future work should determine if polycystin-2 channels function in ER membranes of native tissue and if differential localization confers unique channel properties. These findings present new avenues to study mutant forms of polycystin-2 that cause ADPKD. This method could be used to determine which ADPKD forms of polycystin-2 are gain-of-function or loss-of-function, and perhaps alter channel trafficking to cilia. Ultimately, outcomes from these studies could form a rational basis for polycystin-2 dysregulation in ADPKD and enhance our basic understanding of ciliary ion channel function in cell biology.

## Materials and methods

### Electrophysiology

All electrophysiology reagents used were manufactured by either Sigma Aldrich (St. Louis, MO) or Life Technologies (Carlsbad, CA). . Ciliary ion currents were recorded using borosilicate glass electrodes polished to resistances of 14–18 MΩ using the cilium patch method previously described (*DeCaen et al., 2013*). Holding potentials were −60 mV unless otherwise stated. The pipette standard internal solution (SIS) contained (in mM): 90 NaMES, 10 NaCl, 10 HEPES, 10 Na$_4$-BAPTA (Glycine, N, N'-[1,2-ethanediylbis(oxy-2,1-phenylene)]bis[N-(carboxymethyl)]-,tetrasodium); pH was adjusted to 7.3 using NaOH. Free [$Ca^{2+}$] was estimated using Maxchelator (*Bers et al., 1994*) and adjusted to 100 nM by adding $CaCl_2$. Standard bath solution (SBS) contained 140 NaCl, 10 HEPES, 1.8 $CaCl_2$; pH 7.4. Unless otherwise stated, 'whole cilia' ionic currents were recorded in symmetrical [$Na^+$]. All solutions were osmotically balanced to 295 (±6) mOsm with mannitol. Data were collected using an Axopatch 200B patch clamp amplifier, Digidata 1440A, and pClamp 10 software. Whole-cilium and excised patch currents were digitized at 25 kHz and low-pass filtered at 10 kHz. To accurately measure membrane reversal potential, five current pulses from voltage ramps were averaged. Extra-membrane conditions were changed using a Warner Perfusion Fast-Step (SF-77B) system in which the patched cilia and electrode were held in the perfusate stream. Data were analyzed by Igor Pro 7.00 (Wavemetrics, Lake Oswego, OR). The reversal potential, $E_{rev}$ was used to determine the relative permeability of $K^+$, $Na^+$ and NMDG ($P_X/P_{Na}$) using the following equation:

$$\frac{P_X}{P_{Na}} = \frac{\alpha_{Nae}}{\alpha_{Xe}}\left[\exp(\frac{\Delta E_{rev}}{RT/F})\right]$$

where $E_{rev}$, $\alpha$, R, T and F are the reversal potential, effective activity coefficients for the cations (i, internal and e, external), the universal gas constant, absolute temperature, and the Faraday constant, respectively. The effective activities ($\alpha_x$) were calculated using the following equations:

$$\alpha_x = \gamma_x[X]$$

where $\gamma_x$ is the activity coefficient and [X] is the concentration of the ion. For calculations of membrane permeability, activity coefficients ($\gamma$) were calculated using the Debye-Hückel equation: 0.79, 0.72, 0.30 and 0.24 correspond to $Na^+$, $K^+$, $Ca^{2+}$ and $NMDG^+$, respectively ($\gamma$ for $NMDG^+$ from (*Ng and Barry, 1995*). To determine the relative permeability of $Ca^{2+}$ to $Na^+$, the following equation was used:

$$\frac{P_{Ca}}{P_{Na}} = \frac{\left\{\alpha_{Nai}[\exp(\frac{E_{rev}F}{RT})][\exp(\frac{E_{rev}F}{RT})+1]\right\}}{4\alpha_{Cae}}$$

$E_{rev}$ for each condition was corrected to the measured liquid junction potentials ($-4.4$ to $3.4$ mV).

For the experiments shown in *Figure 7* and *Figure 5—figure supplement 1A*, the internal pipette saline contained 90 mM NaCl, 10 HEPES and 5 $Na_4$-BAPTA and pH was adjusted with NaOH. The extracellular bath solution contained 110 mM X-Cl, 10 HEPES and pH was adjusted with X-OH, where X corresponds to the cation tested ($Na^+$, $K^+$, $Ca^{2+}$, $NMDG^+$). Monovalent-based extracellular solutions contained 0.1 mM EGTA to remove residual divalent cations. The NaMES-based extracellular solution contained: 110 NaMES (sodium methanesulfonate); 10 HEPES; 0.1 EGTA and pH was adjusted with Tris base (tris(hydroxymethyl)aminomethane). For 'on-cilia' single channel recording, the resting membrane potential was neutralized with a high $K^+$ bath solution that contained: 110 KCl, 20 NaCl, 10 HEPES, 1.8 $CaCl_2$, and adjusted to pH 7.4 using KOH. To test the inward single channel conductance, the intracellular pipette solution was replaced with one of the above extracellular solutions. For 'inside-out' cilia recordings, the pipette (extra-ciliary) solution contained SBS and the bath (intraciliary) solution contained 150 NaCl, 10 HEPES, 5 EGTA and free [$Ca^{2+}$] adjusted by adding $CaCl_2$. When a cilium was excised from the cell, the severed end of the cilium commonly re-seals itself, which isolates the intraciliary membrane from the bath and limits the effect of bath applied exchange on the inside of the cilium. To avoid this, excised cilia patches were briefly pressed against the surface of a bead made of Sylgard 184 (Dow Corning) to rupture the cilia at the opposing end. In *Figure 6—figure supplement 1D*, the conductance measurement was made by dividing the concatenated single channel currents against the holding potential. In *Figure 5—figure supplement 1B*, we correct for rundown of the cilia current ($I_{corrected}$) by fitting the control (nominally calcium-free period) inward current to a linear equation (y = mx + b). The difference between $I_{Ca}$ (measured with calcium present) and $I_{corrected}$ was then normalized: [($I_{Ca}$ − $I_{corrected}$)/$I_{corrected}$]× 100. The potency of inward sodium current block was determined by fitting the percent inward current block and calcium concentration relationship to the Hill equation:

$$y = I_{minimum} + \frac{(I_{maximum} - I_{minimum})}{1 + \left[\frac{IC_{50}}{Ca_e}\right]^K}$$

Where $I_{minimum}$ and $I_{maximum}$ are the minimum and maximum response, $IC_{50}$ is the half-maximum inhibition, and K is the slope factor.

## Antibodies, reagents and mice

Mouse monoclonal antibody against GAPDH was from Proteintech. Rabbit polyclonal antibodies against acetylated tubulin (Lys40) was purchased from Cell Signaling Technology (Danvers, MA). Chicken polyclonal antibody against GFP was from Aves Labs (Tigard, OR). Doxycycline and Fluoshield with 1,4-Diazabicyclo [2.2.2] octane were from Sigma-Aldrich; Hoechst 33342 was from Life Technologies. Rabbit polyclonal anti-mouse polycystin 2 (OSP00017W) and rabbit polyclonal anti-mAQP2 (PA5-3800) were from Thermo Fisher (Waltham, MA).

The *Pax8^rtTA; TetO-cre; Pkd1^fl/fl; Pax8^rtTA; TetO-cre; Pkd2^fl/fl* and *Arl13b-EGFP^tg* mice have been previously described(*Ma et al., 2013*)[.41]. *Pax8^rtTA; TetO-cre* and *Pkd1^fl/fl; Pax8^rtTA; TetO-cre; Pkd2^fl/fl* mice were obtained from the Somlo lab (Yale University). *Arl13b-EGFP^tg:cPkd1* or *Arl13b-EGFP^tg: cPkd2* mice were generated by breeding *Pax8^rtTA; TetO-cre; Pkd1^fl/fl* or *Pax8^rtTA; TetO-cre; Pkd2^fl/fl* with *Arl13b-EGFP^tg* mice. The genotype was determined by PCR with the following primers. *Pax8^rtTA* with PCR product ~600 bp: IMR7385-CCATGTCTAGACTGGACAAGA; IMR7386 –CTCCAGGC-CACATATGATTAG. *TetO-Cre* with PCR product ~800 bp: TetO-CMV-5'- GCAGAGCTCGTTTAG TGAAC; Cre-R-TCGACCAGTTTAGTTACCC. *Pkd1^fl/fl* with PCR product ~500 bp (wild type ~300 bp): ND1 Lox 5'-CACAACCACTTCCTGCTTGGTG; ND1 Lox 3'-CCAGCATTCTCGACCCACAAG. *Pkd2^fl/fl* with PCR product ~450 bp (wild type ~300 bp): D2loxF1- GGGTGCTGAAGAGATGGTTC; D2loxR1-TCCACAAAAGCTGCAATGAA. *Arl13b-EGFP^tg* with PCR product ~700 bp (wild type ~400 bp): 83940- TGCAACTCTATATTCAGACTACAG; 84608-GTGGACATAATGGTCCCATTAAGC; Transgene 2562-CATAGAAAAGCCTTGACTTGAGGTTAG. Mice were bred and housed in pathogen-free conditions with access to food and water in the Animal Care Facility. All experimental procedures were approved by the Boston Children's Hospital Animal Care and Use Committee (IUACUC).

## Cell culture

Primary epithelial cells were cultured from dissected kidney collecting ducts of transgenic mice (*Delling et al., 2016*). Inner medullae were removed from the kidney and disassociated using a Dulbecco's phosphate buffered solution (DPBS) containing 2 mg/ml collagenase A and 1 mg/ml hyaluronidase. After mechanical disassociation on ice, medullary cells were cultured in Dulbecco's modified essential medium (DMEM) supplemented with 10% fetal bovine serum (FBS) and 100 units/ml penicillin/100 µg/ml streptomycin. Cilia were patched from cells within 6 days after isolation and within one passage. siRNA knockdown efficiency was monitored for each experiment with a 'scrambled' silencer negative control 1 siRNA (Life Technologies). For generation of *PKD2-GFP* and *GFP-PKD2* stable cell lines, C-terminal or N-terminal GFP-tagged polycystin-2 was generated by subcloning *PKD2* cDNA into a modified pWPXLd vector. Puromycin (2 µg/ml) was used to screen stable cell lines, and these cells were tested for mycoplasma contamination. The *PKD2* (NM_000297) human cDNA ORF Clone was purchased from Origene (Rockville, MD).

## Genomic PCR for pIMCD cells isolated from *Arl13b-EGFP^tg:cPkd1* mice

Genomic DNA was extracted from the pIMCD cells isolated from *Arl13b-EGFP^tg:cPkd1* mice (with and without doxycycline treatment). To genotype the *Pkd1* alleles, primer 1 (P1), 5'-CCGCTGTGTC TCAGTGCCTG −3', and P2, 5'-CAAGAGGGCTTTTCTTGCTG −3', were used to detect the floxed allele (~400 bp) and the *wt* allele (~200 bp); while P1, 5'-CCGCTGTGTCTCAGTGCCTG −3', and P3, 5- ATTGATTCTGCTCCGAGAGG −3', were used to detect the *Deletion* alleles (Deletion of exons 2– 4 of *Pkd1*) of *Del* (~650 bp) and the non-Del (~1.65 kb).

## Immunocytochemistry, confocal microscopy and structured illumination microscopy (SIM)

Cells were fixed with 4% PFA, permeabilized with 0.2% Triton X-100, and blocked by 10% bovine serum albumin in PBS. Cells were labeled with the indicated antibody and secondary fluorescently-labeled IgG (Life Technologies) and Hoechst 33342. Confocal images were obtained using an inverted Olympus FV1000 with a 60x silicon oil immersion, 1.3 N.A. objective. Super resolution images using the SIM method where captured under 100x magnification using the Nikon Structured Illumination Super-Resolution Microscope (N-SIM) with piezo stepping. Confocal and SIM images were further processed with FIJI ImageJ (NIH).

## Immunohistochemistry and cyst parameters

*Arl13b-EGFP^tg:Pax8^rtTA;TetO-cre;Pkd1^fl/fl* and *Arl13b-EGFP^tg:Pax8^rtTA;TetO-cre;Pkd2^fl/fl* mice were induced with 2 mg/ml doxycycline in drinking water supplemented with 3% sucrose for 2 weeks from P28 to P42. Mice were anesthetized and perfused with 4% (wt/vol) paraformaldehyde at 8 weeks and 16 weeks after removal of doxycycline water. Kidneys were harvested and fixed with 4% paraformaldehyde at 4°C overnight, and embedded in paraffin. Sagittal kidney sections were stained with hematoxylin and eosin (H and E) and examined by light microscopy. All kidneys were

**Table 3.** Primers used to detect gene expression using qPCR*.

| Gene, m, *Mus musculus*; H, human | Forward primer 5'−3' | Reverse primer 5'−3' |
| --- | --- | --- |
| m*Pkd1* | CTGGGTGATATTTTGGGACGTAA | GCGTGGCAGTAGTTATCTGCT |
| m*Pkd1-L1* | ATGCCACTCTTGAAGTGAGCA | CCAGGCAGTGTATCTTCTTCCA |
| m*Pkd2* | TACAGCGACCCTCCTTCCC | CCTCTGATGCTCCGACAGATATG |
| m*Pkd2-L1* | CGTGGACATACCATTCCCAGA | ACGGAGAAGTCGATAAAGACCA |
| m*Trpv4* | ATGGCAGATCCTGGTGATGG | GGAACTTCATACGCAGGTTTGG |
| hPKD2 | CGTGCCCCAGCCCAGTC | TTCCAGTACAGCCCATCCAATAAG |

*Athanasia Spandidos, Xiaowei Wang, Huajun Wang, Stefan Dragnev, Tara Thurber and Brian Seed: A comprehensive collection of experimentally validated primers for Polymerase Chain Reaction quantitation of murine transcript abundance. B*MC Genomics* 2008, 9:633.
DOI: https://doi.org/10.7554/eLife.33183.027

photographed under the same magnification. ImageJ analysis software was used to calculate the cyst index (equal to the cumulative area of cysts within the total area of the kidney). For immunofluorescence of acetylated tubulin, a Leica VT1000S vibrating blade microtome was used for sectioning, kidney sections permeabilized with 0.5% TX100/PBS pH 7.4 overnight, and blocked with Block-AidTM solution for 5–8 hr. Sections were washed X3 in PBS, primary antibodies diluted in blocking solution, and sections incubated overnight at 4°C. After slides were washed X3 with PBS, goat anti-chicken/anti-rabbit fluorescent-labeled secondary antibodies were applied at room temperature overnight. Hoechst 33342 nuclear dye was incubated with sections for 1 hr. Sections were washed X3 with PBS, mounted in Fluoshield$_{TM}$ with 1,4-Diazabicyclo [2.2.2] octane and imaged with an inverted Olympus FV1000; silicon oil immersion 60x, 1.3 N.A. objective. Images were further processed and cilia length was measured using Fiji ImageJ (NIH).

### Inhibition and detection of transcripts

Approximately 200,000 primary cells were transfected with 100 pM of siRNAs (ThermoFisher) and 10 μl Lipofectamine RNAiMAX (Life Technologies) in a 9.5 cm$^2$ dish. A list of the siRNAs is described in *Table 1*. At least 48 hr after transfection, half the cells were placed onto glass coverslips for electrophysiology, while the other half were lysed in TRIzol reagent (Ambion) for RNA extraction according to the manufacturer's instructions. RNA was reverse transcribed using the SuperScript reverse transcription kit (ThermoFisher Scientific). Gene-specific primers were designed using Primerbank (http://pga.mgh.harvard.edu/primerbank/)(*Spandidos et al., 2008*). Transcripts were amplified by PCR and expression was visualized by agarose gel electrophoresis. Sequences for gene-specific primers are listed in *Table 3*.

### Statistical analysis

Statistical comparisons were made using two-tailed Student's *t*-tests using OriginPro software (OriginLab, Northampton MA). Experimental values are reported as the mean ± S.E.M. unless otherwise stated. Differences in mean values were considered significant at p<0.05.

## Acknowledgements

We would like to thank the Somlo lab (Yale University) for providing us with the kidney specific, doxycycline-dependent *Pax8$^{rtTA}$; TetO-cre; Pkd1$^{fl/fl}$* (*Shibazaki et al., 2008*) and *Pax8$^{rtTA}$; TetO-cre; Pkd2$^{fl/fl}$* (*Ma et al., 2013*) knockout mice used to cross with the cilia reporter transgenic strain *Arl13b-EGFP$^{tg}$* (*DeCaen et al., 2013*). We thank Nancy and Steven Kleene for the constructive discussions on the electrophysiology data. We thank the Northwestern Nikon Imaging Facility for use of their confocal and super resolution microscopes. We thank members of the Clapham and DeCaen labs for their productive discussions.

## Additional information

### Funding

| Funder | Grant reference number | Author |
|---|---|---|
| Howard Hughes Medical Institute | | David E Clapham |
| National Institutes of Health | Carl W Gottschalk Research Scholar Grant from the American Society of Nephrology | Paul G DeCaen |

The funders had no role in study design, data collection and interpretation, or the decision to submit the work for publication.

### Author contributions

Xiaowen Liu, Conceptualization, Resources, Data curation, Formal analysis, Investigation, Visualization, Methodology, Writing—review and editing, Designed the experiments, Analyzed and interpreted the data, Generated the *Arl13b-EGFP^tg:cPkd1*; *Arl13b-EGFP^tg:cPkd2* animals and the polycystin-2 stably overexpressing cell lines, Conducted, analyzed and interpreted confocal and histology data; Thuy Vien, Writing—review and editing, Conducted the Structured Illumination Microscopy experiments, Assisted with the maintenance of the transgenic animals; Jingjing Duan, Assisted with maintenance of the transgenic animals; Shu-Hsien Sheu, Assisted with the histology of the transgenic animals; Paul G DeCaen, Conceptualization, Resources, Data curation, Formal analysis, Funding acquisition, Investigation, Writing—original draft, Writing—review and editing, Designed the experiments, Analyzed and interpreted data, Performed confocal and electrophysiology experiments, Analyzed and interpreted the results; David E Clapham, Conceptualization, Supervision, Funding acquisition, Investigation, Writing—review and editing

### Author ORCIDs

Xiaowen Liu ⓘ http://orcid.org/0000-0003-4730-2340
Shu-Hsien Sheu ⓘ https://orcid.org/0000-0003-0758-4654
David E Clapham ⓘ http://orcid.org/0000-0002-4459-9428

### Ethics

Animal experimentation: Animals were bred and housed in pathogen-free conditions with access to food and water in the Animal Care Facility. All experimental procedures were approved by the Boston Children's Hospital Animal Care and Use Committee (IUACUC).

### Decision letter and Author response

Decision letter https://doi.org/10.7554/eLife.33183.030
Author response https://doi.org/10.7554/eLife.33183.031

## Additional files

### Supplementary files

• Transparent reporting form
DOI: https://doi.org/10.7554/eLife.33183.028

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
