## [Decision Letter]

Thank you for submitting your article "PKD2 is an essential ion channel subunit in the primary cilium of the renal collecting duct epithelium" for consideration by *eLife*. Your article has been favorably evaluated by Richard Aldrich (Senior Editor) and three reviewers, one of whom is a member of our Board of Reviewing Editors. The following individual involved in review of your submission has agreed to reveal his identity: Ralph Witzgall (Reviewer #3).

The reviewers have discussed the reviews with one another and the Reviewing Editor has drafted this decision to help you prepare a revised submission.

Summary:

This paper by Liu et al. is an important contribution to our understanding of the role of PKD2 (polycystin-2) channels in the cilia of renal epithelial cells. Defects in PKD2 cause some cases of autosomal dominant polycystic kidney disease (ADPKD). Although PKD2 has been studied in various cellular contexts, results have been inconsistent, in part because of the technical difficulty of accessing PKD2 in the primary cilium. In this study, Liu et al. apply sophisticated patch-clamp recording and microscopic techniques to report the first recordings and characterization of PKD2 channels in the primary cilia of primary mouse renal cells. Using Cre-mediated inactivation of the Pkd2 gene, they demonstrate ciliary currents dependent on the presence of PKD2. With a combination of single-channel and whole-cilium currents they show that the channels are permeable to K^+^ and Na^+^ but only sparingly to Ca^2+^ in contrast to previous studies. They are blocked by extracellular Ca^2+^ and sensitized by intracellular Ca^2+^, which may regulate their function in vivo. The PKD2 channels also appear to be active specifically in cilia but not in the cell body, and trafficking to the cilium does not depend on significant expression of PKD1. A surprising finding is that cilia in cystic cells from PKD2 knockout mice are abnormally long. Successful expression of exogenous PKD2-GFP channels in the cilia of HEK cells is also described, offering a useful platform for future studies of mutant PKD2 channels associated with polycystic kidney disease.

Essential revisions:

All three reviewers were enthusiastic about the paper, but felt that the following issues need to be adequately addressed before it can be accepted. Some of the revisions will require additional experiments that can be done with the approaches described in the paper.

1) A central question is whether any PKD1 remains in the cPKD1 cells. While this is mentioned in the Discussion (subsection “Loss of PKD1 does not alter PKD2 ciliary trafficking or PKD2-mediated ciliary currents”, last paragraph), it is hard to judge without better explanations of the methods and interpretation of the results. In particular, Figure 1—figure supplement 2 needs to be more clearly explained. Does the very weak band at 1650 bp in the +dox/Del lane indicate residual, intact Pkd1? Please explain the significance of having the wt (200 bp) and floxed (400 bp) bands both before and after doxycycline treatment. Is the floxed/wt P2 primer site maintained after treatment with doxycycline? A schematic showing the location of the sequences for LoxP, primers, and exons 2-4 would be helpful.

A western blot (the third shown in Figure 1) has staining near one edge of the lane at the expected molecular weight for PKD1. Please comment in the figure legend or Results on whether you think this is residual PKD1 or non-specific background labeling. A lane with lysate from an established PKD1 knockout might help to decide this. Having a trace of PKD1 does not impair the bulk of the paper's findings. It does modestly weaken the claims that PKD1 isn't needed as part of the channel or for PKD2 trafficking.

The same question exists concerning the knockout of PKD2 two weeks after doxycycline treatment. In the first ("2 weeks") western blot of Figure 1, there is a very faint band in the dox-treated lane that is the same size as PKD2. This should be addressed in the results or the figure legend even if it indicates only a tiny amount of PKD2 expression that appears to be gone by two months. Given the genomic data (Figure 1—figure supplement 2) for Pkd1, the possible residual PKD1 (Figure 1), and the residual PKD2 at two weeks (Figure 1), the word "abolished" (subsection “Progressive cyst formation in a new mouse model”, last paragraph) is too strong. PKD2 does appear to be abolished two months after doxycycline – were the recordings with cPkd2 cells all made two months or more after doxycycline treatment?

2) Please explain in the text why whole-cell recording does not measure currents in the cilia (Figure 7, Figure 7—figure supplement 1), and whole-cilia recording does not detect currents from the cell body (Figure 4, Figure 5—figure supplement 1, Figure 7—figure supplement 3). The ability to record currents separately from each of these membrane compartments is key to the conclusions and should be discussed.

3) The ion selectivity of PKD2 channels in this study is similar to but differs from the results obtained by Kleene et al. using the IMCD-3 cell line, especially with regard to Ca^2+^ (P_Ca_/P_Na_ = 0.025 vs. 0.55). Given the number of reports linking PKD2 to Ca^2+^ signaling, this is an important point to nail down. While the authors give several possible reasons for the discrepancy, it is not clear why these (triploidy, divalent contaminants, or channel location) would affect apparent ion selectivity. A more likely explanation may be the different recording modes that were used (single channels in Kleene et al., whole-cilia currents in this study).

There seem to be limitations to both methods, which should be discussed. For the whole-cilia measurements, it seems almost certain that channels other than PKD2 are contributing to the currents, which are evident in the cPKD2 cells after doxycycline (Figure 4). The E_rev_ shift method assumes that background currents are constant throughout the experiment, but some solutions (e.g., isotonic Ca^2+^) may well affect leak and/or seal. Second, even if background current (e.g. Cl^-^ current) is constant, it adds constant terms (P x concentration) to the GHK voltage equation that prevent use of the simple relation (subsection “Electrophysiology”, first equation) to calculate relative permeability. For the single-channel method used by Kleene et al., the measurement was bionic, but reversal potential was estimated from linear extrapolation of the K^+^ currents, which probably biases the E_rev_ to more positive values. It may well be that there is no way to definitely resolve the discrepancies here, but a fuller discussion of these limitations would help the reader to understand the margin of error for each type of measurement. The authors could also comment on why single channel recordings in inside-out cilium patches (as in Figure 6) were not used to measure relative permeabilities.

Finally, the equation used to calculate relative divalent permeability (third equation in the aforementioned subsection) describes a situation with Ca^2+^ on one side and Na^+^ on the other, but it is not clear from the text that this was done. Please clarify.

4) The data clearly show that a whole-cilium current is diminished in dox- treated cPKD2 cells. It is less well established that the single channel described in pIMCD cilia depends on PKD2. It has similar voltage dependence and ionic selectivity as the whole-cilium (PKD2-dependent) current, but identification of the single channels as PKD2 would be much stronger if you could show that they are rare or absent in the cilia of dox-treated cPKD2 cells. (This is especially important for the single 4-pS Ca^2+^ channels seen at very negative potentials, as there is no evidence to show that these are PKD2-dependent.)

5) In HEK cells, a good case is made using immunofluorescence that PKD2-GFP is expressed in the cilia. However, the electrophysiology data do not prove the point. Endogenous PKD2 channels may be present in the cilia, and contributing some or all of the current, and siRNA would reduce that endogenous current as well. It would be useful to show recordings from untransfected HEK cilia if you have them to show that endogenous PKD2 is either absent or at a much lower level, to address this point. Conducting new recordings is not absolutely necessary, but at a minimum a discussion of the issue would be helpful.

6) It is not clear which experiments led to the permeabilities given in the second paragraph of the subsection “PKD2 is primarily a monovalent channel in the cilium”. The P_Na_/P_K_ of 0.4 matches the value given in the first paragraph of the subsection “Ciliary PKD2 preferentially conducts K^+^ and Na^+^ over Ca^2+^ ions”, but the P_Ca_/P_K_ of 0.025 isn't reported anywhere else. The authors should state that number in the Results and describe how it was determined.

7) The expression in the subsection “Electrophysiology” is incorrect; it should be: [(I_control_ – I_Ca_)/I_control_] x 100. Also, the Hill equation should have the I_min_ term in front, not in the numerator: I_min_ + ((I_max_–I_min_)/(1+(IC_50_/X)^k). Also, Minimum and Maximum block rather than I_min_, I_max_, and replace X with Ca_e_ to be consistent with other equations.

8) In Figure 2, cilia became longer after the inactivation of the Pkd2 gene, but in Figure 3 shorter cilia are shown. This is confusing; is this because isolation of primary cells in Figure 3 shortens the cilia? Please explain.

9) The sensitization by intracellular Ca^2+^ was measured at -100 mV. To better understand how this may affect function in intact cells, it is important to consider the voltage dependence of sensitization. How much will Ca^2+^ affect channel activity under physiological conditions of 300-1000 nM Ca^2+^ at -17 mV, the normal resting potential of the cilia?

10) The example of Ca^2+^ block given in Figure 5—figure supplement 1 shows that the current runs down by the end of the experiment, which would lead to an overestimate of the "block" by the higher Ca^2+^ solutions and therefore underestimate the K1/2. Was rundown consistently seen, and how much does this affect the K1/2?

---

## [Author Response]

Essential revisions:All three reviewers were enthusiastic about the paper, but felt that the following issues need to be adequately addressed before it can be accepted. Some of the revisions will require additional experiments that can be done with the approaches described in the paper.1) A central question is whether any PKD1 remains in the cPKD1 cells. While this is mentioned in the Discussion (subsection “Loss of PKD1 does not alter PKD2 ciliary trafficking or PKD2-mediated ciliary currents”, last paragraph), it is hard to judge without better explanations of the methods and interpretation of the results. In particular, Figure 1—figure supplement 2 needs to be more clearly explained. Does the very weak band at 1650 bp in the +dox/Del lane indicate residual, intact Pkd1? Please explain the significance of having the wt (200 bp) and floxed (400 bp) bands both before and after doxycycline treatment. Is the floxed/wt P2 primer site maintained after treatment with doxycycline? A schematic showing the location of the sequences for LoxP, primers, and exons 2-4 would be helpful.A western blot (the third shown in Figure 1) has staining near one edge of the lane at the expected molecular weight for PKD1. Please comment in the figure legend or Results on whether you think this is residual PKD1 or non-specific background labeling. A lane with lysate from an established PKD1 knockout might help to decide this. Having a trace of PKD1 does not impair the bulk of the paper's findings. It does modestly weaken the claims that PKD1 isn't needed as part of the channel or for PKD2 trafficking.The same question exists concerning the knockout of PKD2 two weeks after doxycycline treatment. In the first ("2 weeks") western blot of Figure 1, there is a very faint band in the dox-treated lane that is the same size as PKD2. This should be addressed in the results or the figure legend even if it indicates only a tiny amount of PKD2 expression that appears to be gone by two months. Given the genomic data (Figure 1—figure supplement 2) for Pkd1, the possible residual PKD1 (Figure 1), and the residual PKD2 at two weeks (Figure 1), the word "abolished" (subsection “Progressive cyst formation in a new mouse model”, last paragraph) is too strong. PKD2 does appear to be abolished two months after doxycycline – were the recordings with cPkd2 cells all made two months or more after doxycycline treatment?

This is a fair point and we should have explained more clearly how much PKD1 or PKD2 protein remained after doxycycline-induced Cre expression. The *Pax8^rtTA^; TetO-cre* mice are inducible knockouts of PKD1orPKD2in many, but not all, tubule cells. For example, Cre activity was largely absent in cells from the S3 straight segment. Additionally, non-tubule cells like fibroblasts, interstitial cells, and endothelial cells, also may contaminate the isolated cell preparation. This clarification is added to the third paragraph of the subsection “Loss of polycystin-1 does not alter polycystin-2 ciliary trafficking or polycystin-2 mediated ciliary currents”.

Due to potential impurity of primary cells isolated from *Pax8^rtTA^; TetO-cre; PKD1^fl/fl^; Arl13B-EGFP^tg^* mice, it is possible that the weak band at 1650bp in the +dox/Del lane represents PKD1 from the S3 segment tubule cells or non-tubule cells. The band disappears with a shorter PCR extension time of 20s rather than 45s (the *wt* allele is only 193bp). Thus, the bands around 250bp for the “floxed allele” (lane 2 and 4) without and with doxycycline are nonspecific bands. The interpretation of this data was added to Figure 1—figure supplement 2 legend. The LoxP site between P1 and P2, and the P2 site is removed after doxycycline treatment as shown in Figure 1—figure supplement 2.

In the polycystin western blot in Figure 1, it is possible that a tiny amount of polycystin-2 is represented by the band 2 weeks after doxycycline removal, as previously reported. The band at the left edge (cPKD1, 2 weeks +Doxycycline, first row), appears to be nonspecific, although we cannot rule out polycystin-1 contamination from non-tubule cells. These points are added to the Figure 1 legend.

We agree we cannot state PKD1 and PKD2 are completely removed because there appears to be a trace amount of protein in western blot analysis. We have replaced the word ‘abolished’ with ‘substantially reduced’ or ‘reduced’ (subsection “Progressive cyst formation in a new mouse model”, last paragraph, subsection “Ciliary trafficking and ion channel activity of polycystin-2 are independent of polycystin-1”, last paragraph and subsection “Polycystin-2 is primarily a monovalent channel in the cilium”, third paragraph.

Most important, the cPKD2 currents shown in Figure 4 are much reduced two weeks and two months after removal of doxycycline compared to controls. We did not carry out recordings at times greater than two months after doxycycline treatment.

One point we would like to make: the most sensitive and specific assay we used is patch clamp electrophysiology, which detects protein function to single protein levels. Here we significantly reduced polycystin-1 and still observe the same magnitude of polycystin-2 currents from the discrete primary cilium (Figure 4). If polycystin-1 is necessary – either as a polycystin-2 cilia-chaperone or part of the functional channel complex – its impact on polycystin-2 ion channels should have been detected with the more-sensitive (and functionally relevant) electrophysiological results. It is important to note that the doxycycline-induced reduction of polycystin-1 (or polycystin-2) is sufficient to produce the cystic kidney phenotype in these mice. Thus, the reduced levels –complete or not – are sufficient for their pathophysiological impact on the kidney. This discussion is added to the third paragraph of the subsection “Loss of polycystin-1 does not alter polycystin-2 ciliary trafficking or polycystin-2 mediated ciliary currents”.

2) Please explain in the text why whole-cell recording does not measure currents in the cilia (Figure 7, Figure 7—figure supplement 1), and whole-cilia recording does not detect currents from the cell body (Figure 4, Figure 5—figure supplement 1, Figure 7—figure supplement 3). The ability to record currents separately from each of these membrane compartments is key to the conclusions and should be discussed.

We agree that it is of great importance to record the currents separately from the cell’s plasma membrane and the ciliary membrane, and indeed this is what we have spent several years doing. Whole-cell currents (from pIMCD and HEK cells) capture a distinct set of ion channels from those measured in the whole-cilium configuration. Currents from these membranes can be identified by changing the external cation conditions (compare Figure 7, Figure 7—figure supplement 1 with Figure 4). For example, the pIMCD whole-cell cation currents (Figure 7) consists of an inwardly-rectifying potassium current (Kir), a voltage-gated calcium current (Ca_v_), and an outwardly-rectifying sodium current, whereas the pIMCD whole-cilium current appears to be only one, non-selective outwardly rectifying current, which is dependent on polycystin-2 expression (Figure 4). There is significant electrical resistance between the two compartments (due to the cilium’s ~300 nm radius/several micron length, and restricted volume due to intraciliary, and cilia transition zone proteins) and thus whole-cell vs. whole-cilium ion currents can be separated by breaking into the whole cell, or into the cilium. We also established before, and show again here, that whole-cilium current can be measured by removal of the cilium from the membrane (and resealing of the cilium at its base to form an isolated cilium, or left open to allow intraciliary perfusion), and that these currents are similar to intact, cell attached, whole-cilium recordings. We have added this explanation to the second paragraph of the subsection “Polycystin-2 is primarily a monovalent channel in the cilium”.

Finally, in whole-cell recordings (Figure 7), there were no changes in the currents for the PKD2 knockout cells compared to the control group, which also means that the whole-cilium recordings of polycystin-2 are primarily from the ciliary membrane.

3) The ion selectivity of PKD2 channels in this study is similar to but differs from the results obtained by Kleene et al. using the IMCD-3 cell line, especially with regard to Ca^2+^ (P_Ca_/P_Na_ = 0.025 vs. 0.55). Given the number of reports linking PKD2 to Ca^2+^ signaling, this is an important point to nail down. While the authors give several possible reasons for the discrepancy, it is not clear why these (triploidy, divalent contaminants, or channel location) would affect apparent ion selectivity. A more likely explanation may be the different recording modes that were used (single channels in Kleene et al., whole-cilia currents in this study).There seem to be limitations to both methods, which should be discussed. For the whole-cilia measurements, it seems almost certain that channels other than PKD2 are contributing to the currents, which are evident in the cPKD2 cells after doxycycline (Figure 4). The E_rev_ shift method assumes that background currents are constant throughout the experiment, but some solutions (e.g., isotonic Ca^2+^) may well affect leak and/or seal. Second, even if background current (e.g. Cl^-^ current) is constant, it adds constant terms (P x concentration) to the GHK voltage equation that prevent use of the simple relation (subsection “Electrophysiology”, first equation) to calculate relative permeability. For the single-channel method used by Kleene et al., the measurement was bionic, but reversal potential was estimated from linear extrapolation of the K^+^ currents, which probably biases the E_rev_ to more positive values. It may well be that there is no way to definitely resolve the discrepancies here, but a fuller discussion of these limitations would help the reader to understand the margin of error for each type of measurement. The authors could also comment on why single channel recordings in inside-out cilium patches (as in Figure 6) were not used to measure relative permeabilities.Finally, the equation used to calculate relative divalent permeability (third equation in the aforementioned subsection) describes a situation with Ca^2+^ on one side and Na^+^ on the other, but it is not clear from the text that this was done. Please clarify.

Kleene and Kleene used virus-transformed mouse IMCD-3 cells, which are genetically distinct from the primary collecting duct cells used in this manuscript. They also measured the channel after enveloping the entire cilium and sealing at the plasma membrane junction, which potentially includes contributions from plasma membrane channels. Previously they have reported a 31 pS conductance attributed to TRPM4. Our configuration is unique, where we seal the ciliary membrane exclusively for single channel and whole-cilium measurements. We have not observed the TRPM4 conductance in the pIMCD cilia membrane, but regularly observe a similar conductance (33 pS, among others) in the plasma membrane.

Another difference between our work and the Kleene’s is the ionic recording conditions. In our monovalent conditions, we included 0.1 mM EGTA to reduce divalent (Ca^2+^ and Mg^2+^) contaminants (normally ~10 µM) to nM levels. The Kleene’s consistently added 2 mM Mg^2+^ with varying [Ca^2+^] in both internal and external conditions (Kleene and Kleene, 2017). Thus, their measured monovalent E_rev_ are impacted by these divalent ions, which can confound their interpretation of polycystin-2’s relative permeability. Taken together, membrane type, divalent contaminants, and gene expression differences in our methodology may account for the different selectivities. While there are strengths and weaknesses to both modes, as suggested, we have now estimated the relative permeability of Ca^2+^ based on the extrapolated E_rev_ from our single channel currents (new Figure 5—figure supplement 2). Here we observe ΔE_rev_ = -61 mV in the cilia, supporting our conclusion that Ca^2+^ is less permeable than monovalent ions. Here we assume that the cilioplasm calcium concentration is equal to the measured RPE cilia concentration (580 nM) (Delling et al. 2013), and that the cumulative monovalent concentration in the cilia is equal to cytoplasm (155 mM). When calculated, P_Ca_/P_monovalents_ = 0.04 for the single channel and is consistent with the whole-cilium measurements (P_Ca_/P_Na_ =0.06 and P_Ca_/P_K_ =0.025; method described in our response to point 6), which strengthens our conclusions regarding the low calcium permeability and high selectivity for monovalent ions. Another set of important confirmations are the *Xenopus* oocyte recordings of Pavel et al. (2016), who showed that a gating mutant of PKD2 showed similar selectivity; these recordings were independently reproduced by Mike Sanguinetti in oocytes (personal communication). However, there is no way to fully determine the contributions of each of the differences between the Kleene’s recordings and ours without access to the identical cells and method of recording. This fuller discussion is added to the main text (subsection “Polycystin-2 is primarily a monovalent channel in the cilium”, fourth paragraph). Regarding the conditions and equation in the subsection “Electrophysiology” – yes, calcium is in the external, and sodium was in the internal, solutions (see the aforementioned subsection for intracellular Na^+^ and extracellular Ca^2+^). We have edited the first paragraph of the subsection “Ciliary polycystin-2 preferentially conducts K^+^ and Na^+^ over Ca^2+^ ions”, to be clear.

“Second, even if background current (e.g. Cl^-^ current) is constant, it adds constant terms (P x concentration) to the GHK voltage equation that prevent use of the simple relation in line 586 to calculate relative permeability.”

In Figure 5—figure supplement 1, we tested the P_Cl_^-^ by substituting external chloride anions with methane sulfonate (MES), while keeping sodium constant (110 mM NaCl vs. NaMES) but did not expand or discuss this observation in the original text. Here, there was no difference in reversal potential when NaCl was exchanged for NaMES (ΔE_rev_ = -2 ± 3 mV, Table 2), demonstrating that P_Cl_^-^ is negligible in the cilia membrane. Thus, we can remove P_Cl_ from the suggested GHK expression.

To be clearer on this point, we discussed this observation in the Results and Materials and methods sections (subsection “Ciliary polycystin-2 preferentially conducts K^+^ and Na^+^ over Ca^2+^ ions”, first paragraph; subsection “Electrophysiology”).

4) The data clearly show that a whole-cilium current is diminished in dox- treated cPKD2 cells. It is less well established that the single channel described in pIMCD cilia depends on PKD2. It has similar voltage dependence and ionic selectivity as the whole-cilium (PKD2-dependent) current, but identification of the single channels as PKD2 would be much stronger if you could show that they are rare or absent in the cilia of dox-treated cPKD2 cells. (This is especially important for the single 4-pS Ca^2+^ channels seen at very negative potentials, as there is no evidence to show that these are PKD2-dependent.)

As suggested, we have compared the Ca^2+^ single channels currents from cPKD2 cilia (subsection “Ciliary polycystin-2 preferentially conducts K^+^ and Na^+^ over Ca^2+^ ions”, last paragraph). Both the inward and outward currents were not detected from six cilia after doxycycline treatment (see new figure, Figure 5—figure supplement 2).

5) In HEK cells, a good case is made using immunofluorescence that PKD2-GFP is expressed in the cilia. However, the electrophysiology data do not prove the point. Endogenous PKD2 channels may be present in the cilia, and contributing some or all of the current, and siRNA would reduce that endogenous current as well. It would be useful to show recordings from untransfected HEK cilia if you have them to show that endogenous PKD2 is either absent or at a much lower level, to address this point. Conducting new recordings is not absolutely necessary, but at a minimum a discussion of the issue would be helpful.

This is correct, and we have now discussed this caveat in the last paragraph of the subsection “Polycystin-2 is a ciliary channel but is not constitutively active on the plasma membrane in HEK-293 cells”.

6) It is not clear which experiments led to the permeabilities given in the second paragraph of the subsection “PKD2 is primarily a monovalent channel in the cilium”. The P_Na_/P_K_ of 0.4 matches the value given in the first paragraph of the subsection “Ciliary PKD2 preferentially conducts K^+^ and Na^+^ over Ca^2+^ ions”, but the P_Ca_/P_K_ of 0.025 isn't reported anywhere else. The authors should state that number in the Results and describe how it was determined.

We report P_x_/P_Na_ in the Results (2.4, 1, 0.06 for K^+^, Na^+^, Ca^+^ respectively; subsection “Ciliary polycystin-2 preferentially conducts K^+^ and Na^+^ over Ca^2+^ ions”, first paragraph) as described. We have chosen to keep P_Na_ in the denominator to be consistent with our previous work describing other cilia polycystin channels (Shen et al. 2016; DeCaen et al. *eLife* 2014 and DeCaen et al., 2013). In the third paragraph of the Discussion subsection “Polycystin-2 is primarily a monovalent channel in the cilium”, we convert these values to P_X_/P_K_ (1, 0.4, 0.025) to compare our results to those of Kleene et al. P_Ca_/P_K_ equal to P_Ca_/P_Na_ is divided by P_K_/P_Na_ (P_Ca_/P_K_ =0.06/2.4 = 0.025).

7) The expression in the subsection “Electrophysiology” is incorrect; it should be: [(I_control_ – I_Ca_)/I_control_] x 100. Also, the Hill equation should have the I_min_ term in front, not in the numerator: I_min_ + ((I_max_–I_min_)/(1+(IC_50_/X)^k). Also, Minimum and Maximum block rather than I_min_, I_max_, and replace X with Ca_e_ to be consistent with other equations.

We have made changes (end of subsection “Electrophysiology”) to the equation according to the new method described in point 10 (below).

8) In Figure 2, cilia became longer after the inactivation of the Pkd2 gene, but in Figure 3 shorter cilia are shown. This is confusing; is this because isolation of primary cells in Figure 3 shortens the cilia? Please explain.

In Figure 3, the isolated pIMCD cells were harvested before cysts develop (2 weeks after doxycycline removal), thus there is no difference in the cilia lengths in this figure. This is added in the corresponding figure legend.

9) The sensitization by intracellular Ca^2+^ was measured at -100 mV. To better understand how this may affect function in intact cells, it is important to consider the voltage dependence of sensitization. How much will Ca^2+^ affect channel activity under physiological conditions of 300-1000 nM Ca^2+^ at -17 mV, the normal resting potential of the cilia?

We have added a new figure (Figure 6—figure supplement 1) which describes the calcium dependence with activation, and description in the text (subsection “Ciliary polycystin-2 is sensitized by intraciliary free calcium”).

10) The example of Ca^2+^ block given in Figure 5—figure supplement 1 shows that the current runs down by the end of the experiment, which would lead to an overestimate of the "block" by the higher Ca^2+^ solutions and therefore underestimate the K1/2. Was rundown consistently seen, and how much does this affect the K1/2?

Rundown occurs in most whole-cilium patch experiments, but the rates from patch to patch are inconsistent, thus K_1/2_ cannot be estimated with certainty. The polycystin-2 current from a few cilia ran down extremely fast in 0 mM calcium (>4% of the total current / ramp depolarization) and were not used to generate that data in Figure 5—figure supplement 1. To correct for rundown, we have now normalized the percent block by subtracting the remaining current after calcium addition to the extrapolated control current levels estimated by a linear fit (Figure 5—figure supplement 1). Note that the inward current in this figure returns to the estimated baseline after returning to 0 Ca at the end of the experiment, suggesting that the linear extrapolation currently accurately estimated the amount of current rundown. Using this method (end of subsection “Electrophysiology”), the IC_50_ for external Ca^2+^ block is now = 17 mM (subsection “Ciliary polycystin-2 preferentially conducts K^+^ and Na^+^ over Ca^2+^ ions”, last paragraph).